# The HIV co-receptor CCR5 regulates osteoclast function

Ji-Won Lee [1], Akiyoshi Hoshino[2], Kazuki Inoue[3], Takashi Saitou[4,5], Shunsuke Uehara[6], Yasuhiro Kobayashi[7], Satoshi Ueha[8], Kouji Matsushima[8], Akira Yamaguchi[9], Yuuki Imai [3,10,11] & Tadahiro Iimura [1,3,10]

C–C chemokine receptor 5 (CCR5) is a co-receptor of HIV. Epidemiological findings suggest that the functional loss of CCR5 is correlated with a lower incidence of bone-destructive diseases as well as of HIV transmission. However, it is not clear whether CCR5 is involved in regulation of the function of bone cells, in addition to that of immune cells. Here we show that blockade of CCR5 using specific antibodies impairs human osteoclast function in vitro. *Ccr5*-deficient (*Ccr5*$^{-/-}$) mice presented with dysfunctional osteoclasts and were resistant to osteoporosis induced by receptor activator of nuclear factor kappa-B ligand (RANKL), which triggers osteoporosis independently of inflammatory and immunomodulatory pathways. Furthermore, *Ccr5* deficiency impairs the cellular locomotion and bone-resorption activity of osteoclasts, which is associated with the disarrangement of podosomes and adhesion complex molecules including Pyk2. Overall, the data provides evidence that CCR5 has an essential role in bone-destructive conditions through the functional regulation of osteoclasts.

[1] Division of Bio-Imaging, Proteo-Science Center (PROS), Ehime University, Ehime, 791-0295, Japan. [2] Tokyo Metropolitan Police Hospital, 164-8541 Tokyo, Japan. [3] Advanced Research Support Center (ADRES), Ehime University, 791-0295 Ehime, Japan. [4] Department of Molecular Medicine for Pathogenesis, Graduate School of Medicine, Ehime University, 791-0295 Ehime, Japan. [5] Translational Research Center, Ehime University Hospital, 791-0295 Ehime, Japan. [6] Department of Biochemistry, Matsumoto Dental University, 399-0781 Nagano, Japan. [7] Institute for Oral Science, Matsumoto Dental University, 399-0781 Nagano, Japan. [8] Department of Molecular Preventive Medicine, Graduate School of Medicine, The University of Tokyo, 133-0033 Tokyo, Japan. [9] Oral Health Science Center, Tokyo Dental College, 101-0061 Tokyo, Japan. [10] Artificial Joint Integrated Center, Ehime University Hospital, 791-0295 Ehime, Japan. [11] Division of Integrative Pathophysiology, Proteo-Science Center, Graduate School of Medicine, Ehime University, 791-0295 Ehime, Japan. Correspondence and requests for materials should be addressed to J.-W.L. (email: jwlee@m.ehime-u.ac.jp) or to T.I. (email: iimura@m.ehime-u.ac.jp)

C–C chemokine receptor 5 (CCR5) is mainly expressed on immune cells, including T cells, dendritic cells, monocytes, and macrophages[1]. As CCR5 was found to be a critical co-receptor for macrophage-tropic HIV to achieve its entry into immune cells, a CCR5-antagonist Maraviroc received full FDA approval in 2007 for use in treatment-naive adults with HIV[2]. The lifespan of patients with HIV infection has been significantly increased within the last few decades; simultaneously age-related comorbidities, including the development of bone disease have been found to occur in association with HIV treatment[3–12].

A number of epidemiological studies have demonstrated that non-functional CCR5 is associated not only with lower HIV transmission, but also with the reduced severity rheumatoid arthritis (RA) and/or a lower frequency of RA development[13–17], collectively suggesting that CCR5 is a suitable target for RA therapy. However, some studies have reported contradictory results[18–20]. In the pathogenesis of RA, CCR5 is preferentially expressed in T cells, monocytes, and macrophages, therefore thought to have pro-inflammatory role. However, this hypothesis remains controversial as many background factors in the distinct inflammatory mechanisms and immune responses of individual patients may obscure the association between the loss of the CCR5 function and the development of RA. Accordingly, studies of the potential immunomodulatory and anti-inflammatory effects of the CCR5-antagonist Maraviroc have shown controversial results[21–25].

A recent clinical study reported that Maraviroc was associated with a lower degree of bone loss in the hip and lumbar spine of HIV-infected patients[26]. Some experimental studies demonstrated that CCR5 had direct positive roles in osteoclastogenesis and the communication between osteoclasts and osteoblasts[27–29]. These clinical and basic studies have highlighted another pathway (other than the inflammatory and immunomodulatory pathways) that may explain the beneficial skeletal effects associated with the functional loss of CCR5. On the other hand, Ccr5-deficient mice reportedly showed increased numbers of osteoclasts in a model of alveolar bone resorption induced by orthodontic tooth movement[30]. Mice that genetically lacked Ccl5, which encodes a ligand for CCR5, exhibited increased numbers of osteoclasts and impaired bone formation due to a severe lack of osteoblasts[31]. Thus, the pathophysiological roles of CCR5 in the bone metabolism, which are independent of the inflammatory and immunomodulatory pathways, remain elusive. Furthermore, molecular and cellular function of CCR5-mediated pathway in functional development of osteoclasts have not been well documented.

In the present study, we first investigated the requirement of CCR5 in the differentiation of cultured human and mouse osteoclasts. We next analyzed the bone phenotypes of Ccr5-deficient mice in a model of pathological bone destruction, which was induced by the administration of RANKL. Our findings demonstrate that CCR5 is required for the functional cellular architecture of osteoclasts through regulating integrin- and chemokine-mediated pathways, thereby elucidating the direct association between loss of CCR5 and resistance to bone loss in mice.

## Results

### The requirement of CCR5 for human osteoclast function. We first investigated the necessity of the CCR5 during the differentiation of human osteoclasts in vitro by administering a CCR5-neutralizing antibody (anti-hCCR5 neuAb) (Fig. 1a). The blockade of CCR5 for the duration of the entire differentiation process on days 2–6 (D2-6) clearly inhibited the formation of actin rings and resorption pits on dentin slices in a dose-dependent manner

(Fig. 1b). Pulse incubation with anti-hCCR5 neuAb for five distinct periods (days 2–6 [D2-6], 0-1 [D0-1], 0-2 [D0-2], 2–4 [D2-4], and 4–6 [D4-6]) demonstrated the temporal requirement of CCR5 for tartrate-resistant acid phosphatase (TRAP) activity in relatively late stages (D2-4 and D4-6) (Fig. 1c), which was also morphologically confirmed (Fig. 1d). Pertussis toxin (PTX), an inhibitor of Gαi and thus a pan-inhibitor of chemokine receptors, effectively reduced these osteoclastic parameters (Fig. 1b–d). We also noticed that the osteoclasts cultured on a glass-bottomed dish in incubation with anti-hCCR5 neuAb (D2-6) often failed to form actin rings (Fig. 1e, f, left panels). We then took advantages of structured illumination microscopy (SIM) to resolve the actin-enriched podosomes constituting the actin rings (Fig. 1e–h). The blockade of CCR5 (D2-6) significantly inhibited both the formation and assembly of podosomes (Fig. 1f–h). The actin stress fibers inside the podosome belts were also disrupted by the CCR5 blockade (Fig. 1f, right panels). This CCR5 blockade at optimal doses (1 µg mL$^{-1}$ and 10 µg mL$^{-1}$) did not affect the proliferation of human osteoclast precursors (cells prior to RANKL stimulation) (Supplementary Fig. 1A).

We next used U-937 cells[32] for their better gene transfer rate and more feasible expansion of cell population than those of normal human osteoclast precursors, which enabled us to conduct statistical analyses of osteoclastic markers (Supplementary Fig. 1B, C). The CCR5-knocked-down cells treated with the anti-hCCR5 neuAb (D2-6) did not have any additional inhibitory effect on the TRAP activity, thus confirming the specificity of anti-hCCR5 neuAb. Furthermore, the CCR5-antagonist Maraviroc or the anti-hCCR5 neuAb significantly reduced the expression levels of CTSK, NFATC1, and ACP5 in U-937 cells in a dose-dependent manner (Supplementary Fig. 1C).

We next tested whether the phenotypic effects on actin ring formation were reversible or irreversible after the depletion of the anti-hCCR5 neuAb (Supplementary Fig. 1D, E). We cultured human osteoclasts with control antibodies, or anti-hCCR5 neuAb on dentin slices for days 4–6 (D4-6), and then treated them with cold PBS for 10 min that caused obvious shrinkage of the formed actin rings (middle panels in Supplementary Fig. 1D). Interestingly, returning these cell cultures to the normal differentiation medium clearly allowed the cells to reform actin rings in all three conditions (right panels in Supplementary Fig. 1D). SIM confirmed the re-assembly of podosomes in osteoclasts cultured on a glass-bottomed dish (Supplementary Fig. 1E).

The blockade of CCR5 in the differentiation of human osteoblasts (hOBs) from mesenchymal stromal cells (hMSCs) did not cause any significant changes in the mineralization or expression levels of RUNX2, SP7, TNFSF11, and ALPL (Supplementary Fig. 2A, B).

### Disarrangements of podosomes in Ccr5$^{-/-}$ osteoclasts. To further elucidate the role of CCR5 in osteoclastogenesis, we cultured bone marrow cells (BMCs) obtained from Ccr5-deficient (Ccr5$^{-/-}$) mice and their wild-type littermates (the cell culture schedule is shown in Fig. 2a). The Ccr5$^{-/-}$ osteoclasts cultured on a glass-bottomed dish were significantly larger in size and showed defective actin ring formation (Supplementary Fig. 3A). The cellular proliferation of Ccr5$^{-/-}$ bone marrow macrophages (BMM) was significantly increased on day 4 after the initial incubation with macrophage-colony stimulating factor (M-CSF) (Supplementary Fig. 3B, left), whereas the TRAP activity of the Ccr5$^{-/-}$ osteoclasts was comparable to that of wild-type cells (Supplementary Fig. 3B, right). The actin rings in Ccr5$^{-/-}$ osteoclasts on dentin slices, however, were significantly disrupted (Fig. 2b). Our scoring indicated that the actin rings in Ccr5$^{-/-}$ osteoclasts fractioned and significantly reduced in size, thus

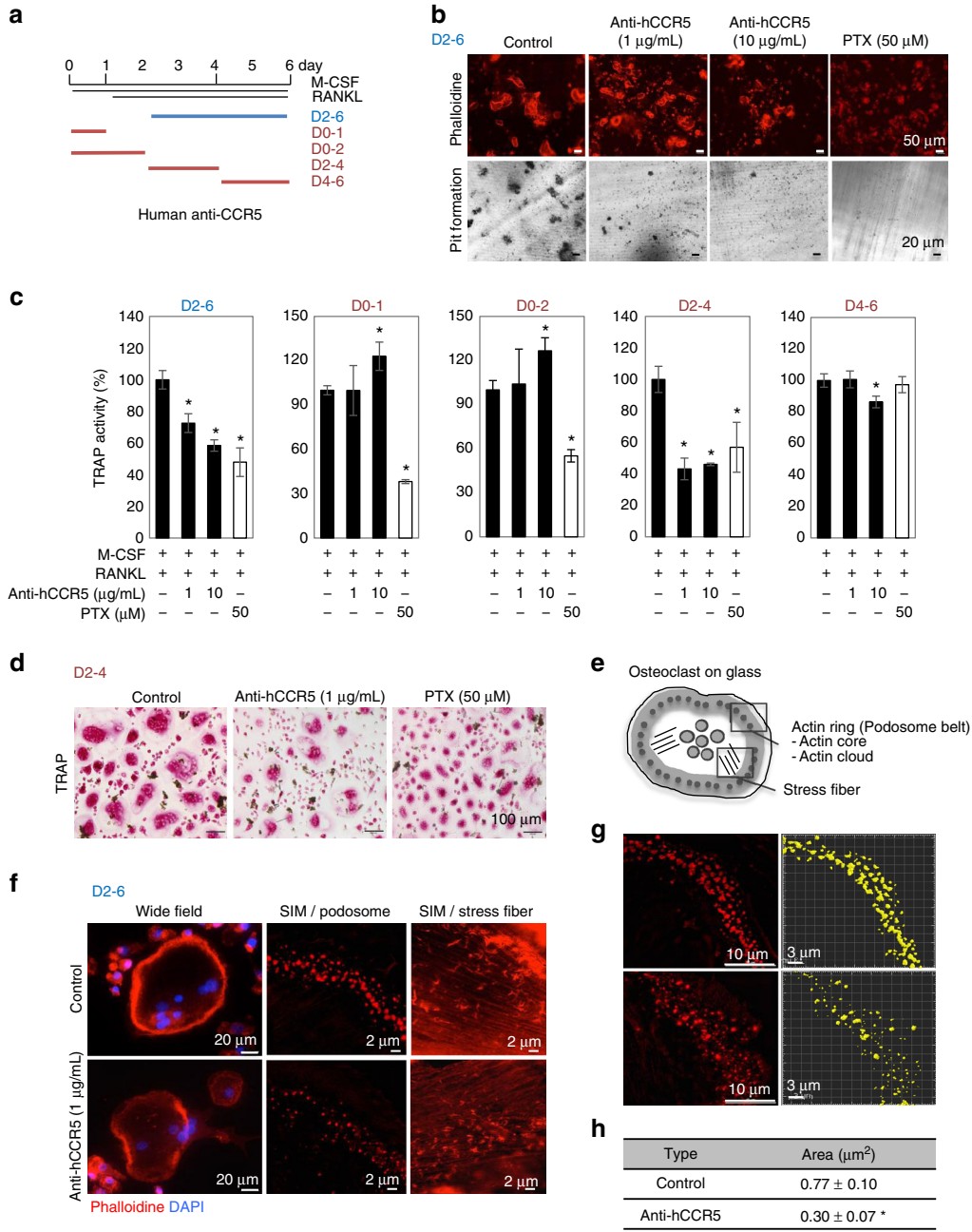

**Fig. 1** Blockade of CCR5 in cultured human osteoclasts. **a** A schematic illustration of the time schedule for the continuous or intermittent exposure of cultured human osteoclasts to anti-hCCR5 neuAbs. **b** The human osteoclasts were cultured in the presence of anti-hCCR5 neuAbs (1 μg mL$^{-1}$, 10 μg mL$^{-1}$) or pertusiss toxin (PTX) (50 μM) for day 2–6. Mature osteoclasts were visualized by staining F-actin with phalloidin-AlexaFluor568 (scale bars, 50 μm, $n = 4$) and a pit formation assay was performed (scale bars, 20 μm, $n = 4$). **c** The human osteoclasts were cultured with anti-hCCR5 neuAbs in a distinct, intermittent manner, as shown in **a**. The TRAP activity levels were scored and compared ($n = 5$). **d** Cells were stained to show TRAP activity (scale bars, 100 μm, $n = 5$). **e** A schematic drawing shows the spatial locations of the podosome belt and stress fibers in a mature osteoclast cultured on a glass-bottomed dish. **f** Standard wide-field fluorescence images (left panels) and super-resolution images of podosome belts (middle panels) and stress fibers (right panels) stained with phalloidin-AlexaFluor568 (scale bars, 20 μm, $n = 3$). Super-resolution images were captured by SIM (magnification ×100 (objective lens); scale bars, 2 μm, $n = 3$). **g, h** After obtaining three-dimensional SIM images (scale bars, 10 μm), the size of the actin-enriched podosome cores (actin core spots) were quantified by a surface rendering-based analysis (scale bar, 3 μm, $n = 6$) and were statistically compared. *$P < 0.05$ by Student's $t$-test. The data shown as the mean ± SD

significantly increased their number compared to those in wild-type cells. Furthermore, the $Ccr5^{-/-}$ osteoclasts significantly reduced pit formation and transcriptional levels of $Mmp3$ and $Mmp13$ (Fig. 2c, d).

SIM further showed that vinculin molecules surround podosomes and are well associated with each other in wild-type cells.

However, in $Ccr5^{-/-}$ osteoclasts, the distribution of vinculin molecules was dispersed and dissociated with podosomes (Fig. 2e, right panels). These findings demonstrated that $Ccr5$ deficiency resulted in the reduced bone-resorption activity of osteoclasts in association with failure in the spatial arrangement of podosomes and a focal adhesion-related molecule.

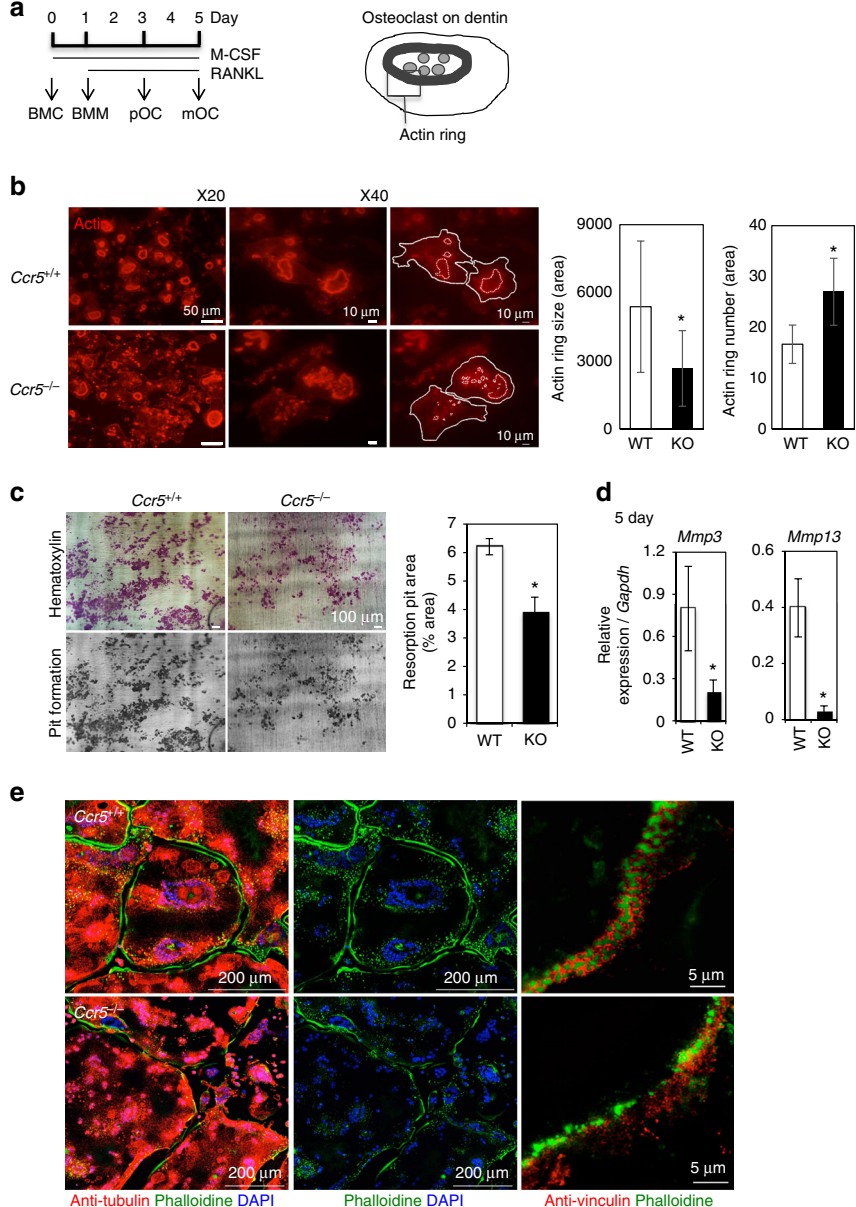

**Fig. 2** The impaired adhesion structures in *Ccr5*-deficient (*Ccr5*⁻/⁻) osteoclasts. **a** (left) A schematic illustration of the time schedule for mouse osteoclast culture from BMCs . (right) A schematic drawing shows the spatial locations of the actin ring (podosome belt) in a mature osteoclast cultured on a dentin slice. **b** Actin ring formation assays of wild-type (*Ccr5*⁺/⁺) and *Ccr5*⁻/⁻ osteoclasts on dentin slices. Cells were stained with phalloidin-AlexaFluor568 (shown in red). The magnification of the upper and lower panels was ×20 (scale bars, 50 μm) and ×40 (scale bars, 10 μm), respectively (*n* = 5). The size and number of actin ring per area were scored and statistically compared. **c** Resorption pit assays. After matured, the osteoclasts were removed from the dentin slices and were subsequently stained with hematoxylin to visualize the resorption pits. Color images (upper panels) and gray-scale images (lower panels) are shown (scale bars, 100 μm). The resorption pit areas (%) on the gray-scale images were scored and statistically compared (*n* = 5). **d** The relative mRNA expression levels of *Mmp3* and *Mmp13* were measured by a real-time Q-PCR and statistically compared on day 5. The data shown as the mean ± SD (*n* = 5). **e** The formation of actin rings (podosome belts) in wild-type and *Ccr5*⁻/⁻ mature osteoclasts was visualized by enhanced resolution confocal imaging (left 4 images, scale bars, 200 μm). The cells were subjected to immunohistochemical staining with anti-tubulin antibodies (shown in red), and were concomitantly stained with phalloidin-AlexaFluor488 and DAPI to visualize the actin rings (shown in green) and nuclei (in blue), respectively (*n* = 4). SIM images (right two panels) demonstrate the assembly of actin-enriched podosome cores and vinculins in wild-type and *Ccr5*⁻/⁻ mature osteoclasts. The cells were subjected to immunohistochemical staining using anti-vinculin antibodies (shown in red), and were concomitantly stained with phalloidin-AlexaFluor488 to visualize the actin rings (shown in green, scale bars, 5 μm, *n* = 4). *$P < 0.05$ by Student's *t*-test. All data are shown as the mean ± SD

**Disarrangement of adhesion molecules in *Ccr5*⁻/⁻ osteoclasts.** The impaired podosome assembly in *Ccr5*⁻/⁻ osteoclasts prompted us to analyze the movements of osteoclasts derived from *Ccr5*⁻/⁻ and wild-type mice on culture dishes by time-lapse microscopy. Figure 3a shows representative time-lapse images showing the cellular movements of the *Ccr5*⁻/⁻ and wild-type osteoclasts (see also Supplementary Movies 1 and 2). The contraction and extension areas of the cellular periphery in 10 min interval were scored (marked in red and green, respectively, in Fig. 3a). The cell deformation index (CDI) showed that the

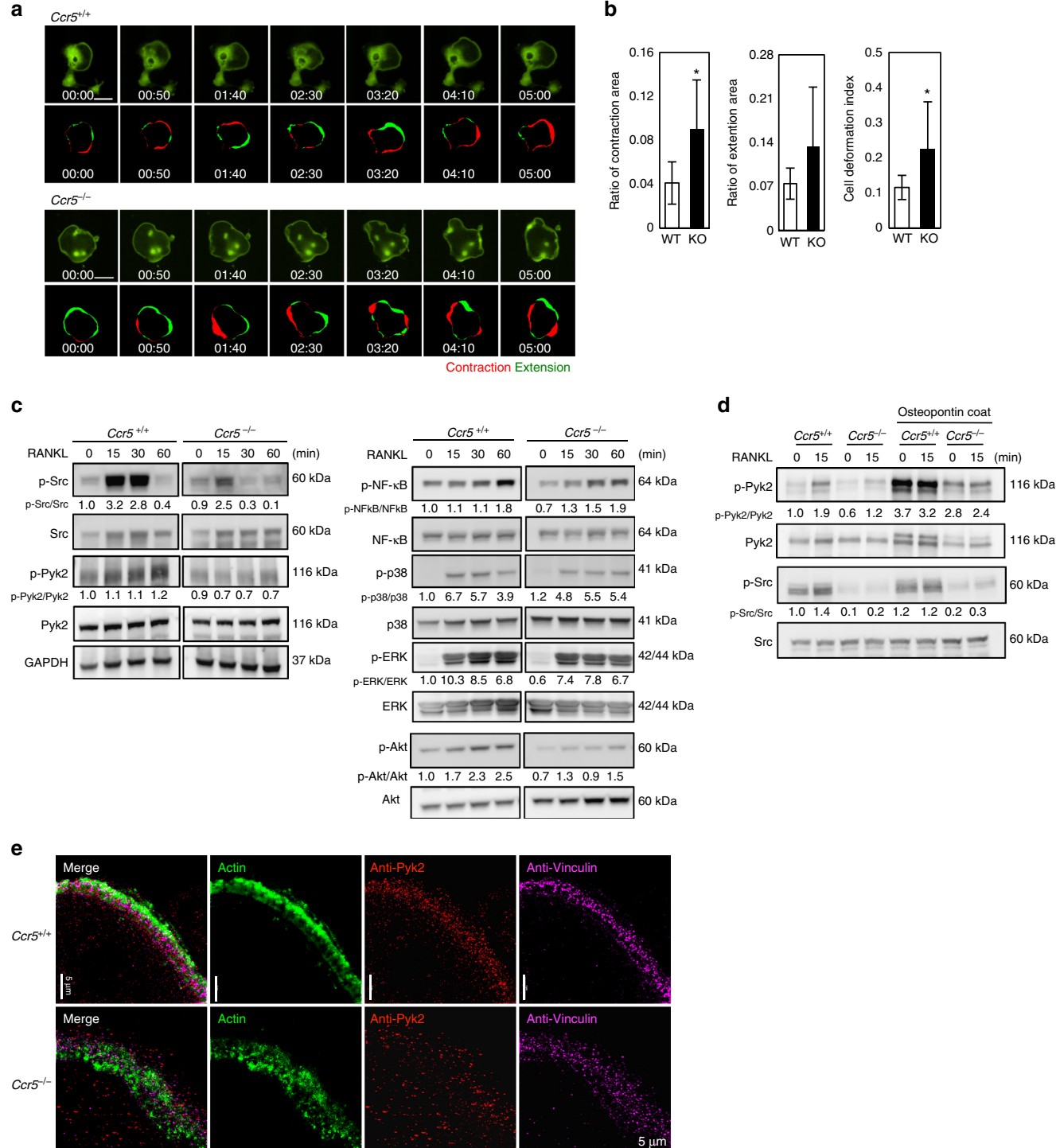

**Fig. 3** The locomotive and adhesion defects in *Ccr5*-deficient (*Ccr5*$^{-/-}$) osteoclasts. **a** Time-lapse microscopy frames show the locomotion of wild-type (*Ccr5*$^{+/+}$) and *Ccr5*$^{-/-}$ osteoclasts expressing GFP (scale bars, 100 μm). The patterns of cellular motility were scored with the ratios of the contraction (shown in red) and extension (in green) areas. The contraction and extension areas were scored by 10 min interval time-lapse images. Time-laps movies are shown in Supplementary Movies 1 and 2. GFP-expressing multi-nucleated cells from wild-type and *Ccr5*$^{-/-}$ mice were analyzed ($n = 9$ and 6, respectively). Cells showing the parameters closest to the mean values are shown. **b** The ratios of the contraction and extension areas, and the cell deformation index (CDI) were analyzed and statistically compared. *$P < 0.05$ by Student's *t*-test. The data shown as the mean ± SD ($n = 6-7$). **c**, **d** BMCs obtained from wild-type and *Ccr5*$^{-/-}$ cells were subjected to immunoblotting of phosphorylated Src, Pyk2, NF-κB, p38 and ERK, and Akt. Prior to the analyses, the cells were stimulated by RANKL for the indicated time. The immunoblotting data were replicated more than three times. **e** Three-dimensional (3D)-SIM images demonstrate the assembly of actin-enriched podosome cores, Pyk2, and vinculins in mature wild-type and *Ccr5*$^{-/-}$ osteoclasts. The cells were subjected to immunohistochemical staining using anti-Pyk2 antibodies (shown in red) and anti-vinculin antibodies (shown in pink), and were concomitantly stained with phalloidin-AlexaFluor488 to visualize the actin rings (shown in green). Maximum intensity projection images of 3D-SIM optical slices are shown (scale bars, 5 μm, $n = 4$). Reconstructed 3D-SIM images are shown in Supplementary Movies 3 and 4

cellular locomotion of $Ccr5^{-/-}$ osteoclasts was significantly increased in comparison to wild-type cells (Fig. 3b), suggesting the unstable attachment to extracellular matrix.

Pyk2 is one of the main kinase involved in the adhesion complex of osteoclasts in association with Src[33–39]. Thus, we next investigated phosphorylations of Src and Pyk2 in response to RANKL stimulation by immunoblotting (Fig. 3c). In wild-type cells, the phosphorylation of Src and Pyk2 were promptly activated by RANKL stimulation. In $Ccr5^{-/-}$ osteoclasts, however, these phenomena were markedly reduced while the differences in the phosphorylation of NF-κB, p38, and ERK in $Ccr5^{-/-}$ osteoclasts were comparable to wild-type cells. Phosphorylation of AKT after RANKL stimuli was also reduced in $Ccr5^{-/-}$ osteoclasts[40].

The phosphorylation of Pyk2 in osteoclasts is activated by integrin ligands as well as by RANKL stimulation. When osteopontin, a major ligand for integrins in osteoclasts, was coated on culture dishes, RANKL stimulation induced the phosphorylation of Src and markedly enhanced the level of phosphorylated Pyk2 in wild-type cells; however, these activations were also hampered in $Ccr5^{-/-}$ osteoclasts (Fig. 3d). SIM demonstrated that Pyk2 and vinculin were highly associated with actin-enriched podosomes in wild-type osteoclasts, indicating the proper arrangement of the adhesion complex of osteoclasts, whereas $Ccr5^{-/-}$ osteoclasts exhibited disarrangement of these molecules and podosomes (Fig. 3e; see also Supplementary Movies 3 and 4). Collectively, these findings suggested that $Ccr5$ deficiency affected the cellular locomotion of the osteoclasts associated with the spatially impaired arrangement of podosomes and adhesion complex that involves Pyk2.

**CCL5 enhances integrin- and chemokine-mediated signaling.** We confirmed expression of CCR5 in osteoclasts both in vivo and vitro. Double immunofluorescence staining of wild-type bone section showed that CCR5 was predominantly expressed in multinuclear Cathepsin K-positive osteoclasts (Fig. 4a). Consistently, in cultured wild-type osteoclasts, the $Ccr5$ expression increased in the later stages of osteoclast differentiation, and was relatively higher than other C–C chemokine receptors in entire culture period (Fig. 4b; Supplementary Fig. 4A). Our previous study also reported that CCL5 and CCL9 (ligands for CCR1 and CCR5) were endogenously expressed by osteoclasts and were required for osteoclast differentiation in vitro[41]. $Ccl5$ was highly detectable throughout the culture period, whereas $Ccl9$ always remained at a basal level (Supplementary Fig. 4B). This finding suggested that CCL5 was more functional ligand than CCL9 in osteoclast differentiation at least in vitro.

We next conducted a population analysis of RANK-positive cells induced by M-CSF and RANKL stimulation (Supplementary Fig. 5A, B). After 2 days of culturing, we observed that G3-gated cells (CX3CR1$^{low}$ and CD115$^{high}$) showed a cell population that exhibited the highest expression of RANK. These populations derived from the two distinct genotypes did not show any significant changes, suggesting that the initial differentiation potential of wild-type and $Ccr5^{-/-}$ cells into osteoclasts was comparable. Live-cell imaging analysis and a small pore-sized filter chamber assay both demonstrated that the cellular locomotion and the velocity of migration toward M-CSF were comparable in BMCs with the two distinct genotypes (Supplementary Fig. 5C, D; see also Supplementary Movies 5 and 6). These findings collectively demonstrated that the phenotypic requirement of $Ccr5$ tended to be observed in the later functional stages of osteoclast differentiation.

Treatment with recombinant mouse CCL5 (rmCCL5) significantly augmented the number of wild-type-derived osteoclasts

but not $Ccr5^{-/-}$ cells (Fig. 4c, d), indicating that CCL5 enhanced RANKL-induced osteoclastogenesis through CCR5. The incubation with rmCCL5 also increased the numbers of actin rings and resorption pits of cultured wild-type osteoclasts on dentine slices in a dose-dependent manner (Supplementary Fig. 6A). We further investigated the roles of the CCL5 in the crosstalk with the RANKL-induced downstream signals. The rmCCL5 stimulation induced the phosphorylation of FAK, even at 0 min (prior to RANKL treatment) (Fig. 4e). This did not occur after RANKL stimulation without rmCCL5. This elevated level of the phosphorylated FAK by the pre-incubation with rmCCL5 lasted in 30 min after RANKL stimulation. This pre-incubation with rmCCL5 also enhanced the phosphorylation of Pyk2 prior to RANKL treatment. RANKL stimulation induced the phosphorylations of Src at 15–30 min and Pyk2 at 15–60 min after the stimulation. To confirm the induced phosphorylation of Src by RANKL or the combination of rmCCL5 and RANKL, we conducted immunoprecipitation with phospho-tyrosine antibodies. The level of phosphorylated Src following the combination treatment with RANKL and CCL5 was markedly increased in comparison to that with RANKL alone (Fig. 4f). Inoculation with rmCCL5 significantly activated Rac, but not Rho (Fig. 4g).

To investigate the changes in the transcriptional signatures induced by CCL5 in osteoclastogenesis, cultured osteoclasts at pOC stage (see Fig. 2a) were incubated with or without rmCCL5 for 2 days, and then subjected for RNA-sequencing. Genes with a log2 (fold-change) value of ≥0.5 and an adjusted $P$-value of <0.05 were tested for enrichment using the Kyoto Encyclopedia of Genes and Genomes (KEGG) pathway database (Fig. 4h–j). A gene ontology analysis demonstrated that the greatest number of genes that were upregulated by CCL5 in osteoclastogenesis were found in pathways related to lysosomes, ECM-receptor interaction, focal adhesion, cell adhesion, osteoclast differentiation, and the chemokine signaling pathway (Fig. 4i, j). Interestingly, the pathway analysis demonstrated that transcriptionally upregulated molecules were involved in the integrin- and chemokine-mediated pathways (Supplementary Fig. 7).

**Rac or Rho corrects the dysfunction of $Ccr5^{-/-}$ osteoclasts.** On the basis of our pathway analysis, we examined the activities of small GTPases as intracellular signaling molecules that are involved both in focal adhesion- and chemokine-mediated signals in $Ccr5^{-/-}$ osteoclasts[42,43]. In $Ccr5^{-/-}$ osteoclasts, the levels of Vav3 and the phosphorylated forms of FAK (revealed by immunoblotting) were markedly downregulated in comparison to wild-type cells (Fig. 5a). We next investigated small GTPases such as Rac, Rho, and Cdc42-as possible downstream targets of this signaling[43] (Fig. 5b). The active forms of Rac, Rho, and Cdc42 were upregulated in wild-type cells in the pre-osteoclast (pOC) stage, however, in $Ccr5^{-/-}$ cells, the active forms of Rac, Rho but not Cdc4 were significantly suppressed in this stage.

To test the relevance of these suppressed functions of Rac and/ or Rho for the dysfunctional phenotype of $Ccr5^{-/-}$ osteoclasts, we expressed constitutively active forms of Rac and Rho (Rac-CA and Rho-CA, respectively) in $Ccr5^{-/-}$ osteoclasts by adenovirus-mediated gene transfer. A numerical assay of adhesion ring formation based on the immunofluorescence of vinculin showed that the expression of Rac-CA or Rho-CA in $Ccr5^{-/-}$ osteoclasts led to recoveries of their impaired peripheral outlines (Fig. 5c). The pit formation assay revealed that this parameter, which were impaired in $Ccr5^{-/-}$ osteoclasts, were significantly upregulated to the levels in wild-type cells by the expression of either Rac-CA or Rho-CA (Fig. 5d). The reduced mRNA expression levels of $Itg\alpha V$ in $Ccr5^{-/-}$ osteoclasts were significantly increased to the levels of wild-type cells by the expression of either Rac-CA or Rho-CA

(Fig. 5e). The reduced expression of *Mmp3* and *Mmp13* in *Ccr5*$^{-/-}$ osteoclasts was corrected by Rho-CA and Rac-CA, respectively (Fig. 5e). These experimental findings indicated that the dysfunction of *Ccr5*-deficient osteoclasts was sufficiently corrected by the functional recovery of Rac or Rho, though this correction by either small GTPase occurred in a distinct manner to a certain extent.

**$Ccr5^{-/-}$ mice were tolerant to the RANKL-induced bone loss.** To investigate the possible roles of CCR5 in the physiological and

pathological bone-destructive conditions in bone metabolism, we took advantages of a RANKL-induced bone loss model (Fig. 6). In the vehicle (PBS) injection groups of *Ccr5*$^{-/-}$ and wild-type mice, μCT-based parameters such as the bone mineral density (BMD), bone volume/tissue volume (BV/TV), and trabecular connective density (Conn-Dens.) showed no significant changes (Fig. 6a, b), while the trabecular number (Tb.N) was significantly reduced in *Ccr5*$^{-/-}$ mice. These parameters in the vehicle injection groups of two genotypes did not exhibit overt phenotypes in bone architecture of *Ccr5*$^{-/-}$ mice.

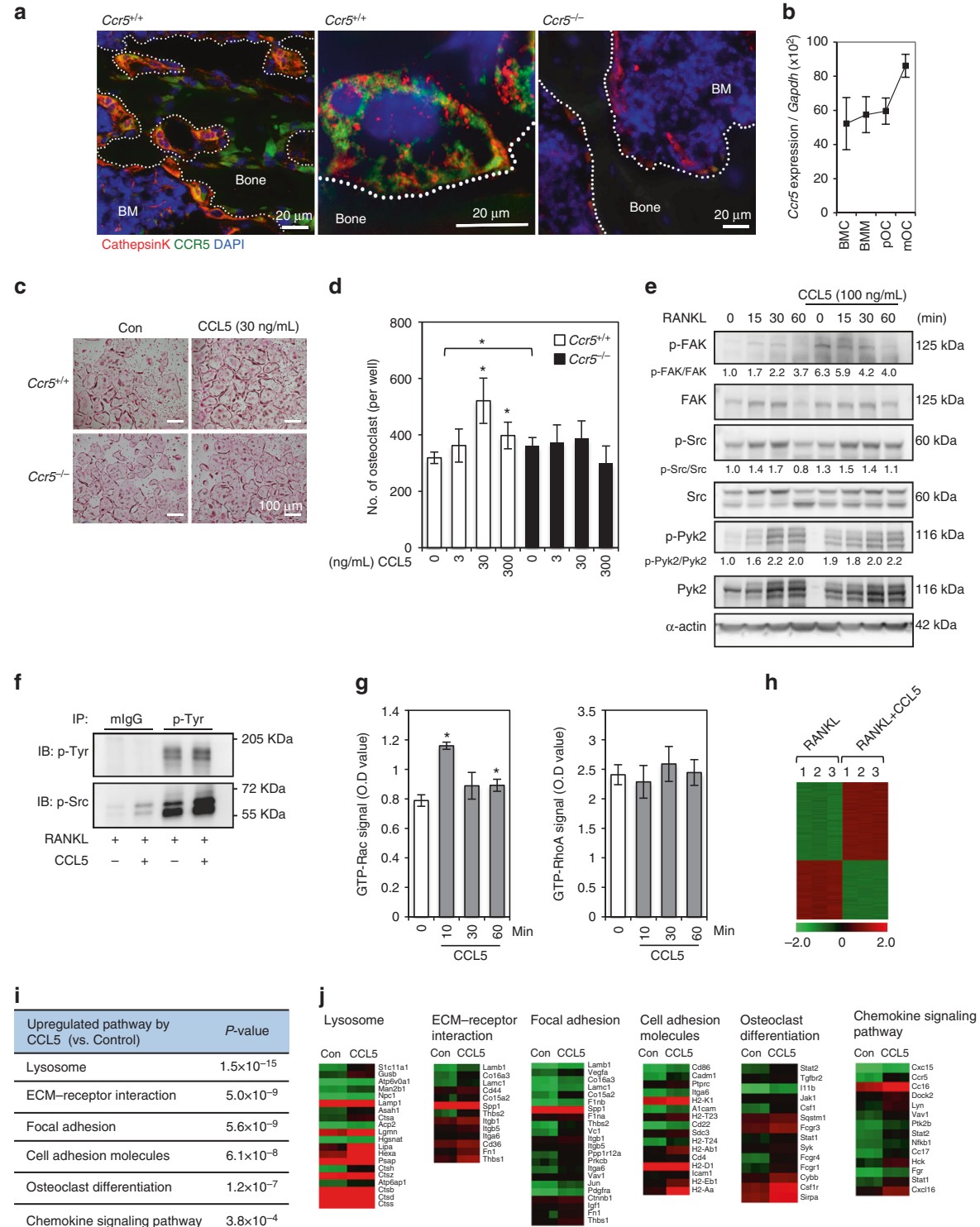

The administration of the recombinant soluble RANKL (sRANKL; 2 mg/kg, i.p.) to wild-type induced a significant reduction in the µCT-based parameters such as BMD, BV/TV, Tb.N, and Conn-Dens., in comparison to the vehicle-injected wild-type group (Fig. 6a, b). The serum levels of OPG, TRAP, and CTX were significantly augmented in the sRANKL-injected wild-type mice, whereas the endogenous levels of M-CSF, RANKL, and osteocalcin (OCN) did not differ to a statistically significant extent, thus validating the bone loss model (Fig. 6c; Supplementary Fig. 8A). In contrast, the administration of sRANKL to $Ccr5^{-/-}$ mice did not alter these µCT-based parameters or the serum levels of OPG or CTX in comparison to the vehicle-injected $Ccr5^{-/-}$ group. The TRAP level was significantly augmented by the injection of RANKL, but to less extent compared to that in their wild-type littermates. These analyses indicated that $Ccr5^{-/-}$ mice were more resistant to sRANKL-induced osteoporosis than their wild-type littermates.

Histomorphometric analyses (Fig. 6d, e; Supplementary Fig. 8B) showed that the $Ccr5^{-/-}$ mice in the vehicle control group showed increases in the number of osteoclasts (N.Oc), the osteoclast surface per bone surface (Oc.S/BS), the number of osteoclasts per bone perimeter (N.Oc/B.Pm), and a decreased number of osteoclasts per osteoclast perimeter (N.Oc/Oc.Pm) in comparison to their wild-type littermates in the vehicle control group (Fig. 6e). Consistently, the numbers of TRAP-positive osteoclasts in histological sections of $Ccr5^{-/-}$ trabecular femoral bone were markedly increased in comparison to the wild-type femoral bone (Fig. 6d). Furthermore, a magnified view showed flattened osteoclasts that covered a relatively wider trabecular bone surface in the $Ccr5^{-/-}$ femoral bone in comparison to the wild-type femoral bone. Histomorphometric parameters of osteoblasts were comparable between the vehicle-injected groups of the two genotypes (Supplementary Fig. 8B). These histomorphometric data in the vehicle-injected groups of two genotypes unveiled covert osteoclast dysfunction in $Ccr5^{-/-}$ mice.

The administration of sRANKL to wild-type mice resulted in a significant increase in their N.Oc, Oc.S/BS, and N.Oc/B.Pm values in comparison to the wild-type vehicle control group, confirming that sRANKL successfully induced bone loss. However, the administration of sRANKL to $Ccr5^{-/-}$ mice did not affect these parameters. These data demonstrated that CCR5 had an essential role in the osteoclast function in vivo, and suggested that CCR5 has a more evident role in bone-destructive conditions.

**Critical roles of CCL5 in bone mass regulation**. Our in vitro analyses suggested that CCL5 is a major functional endogenous ligand for CCR5 and CCR1. CCL3 has been reported to be involved in models of pathological bone-destructive disease[28,29,44–46]. Thus, we investigated the serum levels of CCL3 and CCL5 in physiological conditions (Fig. 7a). The serum concentration of CCL5 was ~60 ng mL$^{-1}$, whereas the level of CCL3 was undetectable by an ELISA-based method. The immunofluorescence of CCL5 in the bone marrow of the 8-week-old mouse tibia exhibited relatively intense signals in osteoblasts and osteoclasts covering the trabeculae, endothelial cells, and stromal cells (Supplementary Fig. 6B). To ascertain whether CCL5 exerts a functional role in the bone in vivo, the antibodies specific to mouse CCL5 (CCL5 neuAb) were intraperitoneally injected (once a week for 2 weeks) into 6-week-old C57BL/6J male mice (Fig. 7b). The µCT images of CCL5 neuAb-injected mice showed thickened cortical and trabecular bones in comparison to control mice (Fig. 7c). In fact, µCT-based parameters such as BMD, BV/TV, Conn-dens., and Tb.N showed a significant increase in the bone architecture of CCL5 neuAb-injected mice (Fig. 7d). The histological analysis showed that the numbers of TRAP-positive osteoclasts in CCL5 neuAb-injected mice were markedly reduced in comparison to control mice (Fig. 7e). In CCL5 neuAb-injected bones, these cells were obviously smaller in size and occasionally exhibited a flattened shape. The histomorphometric parameters of osteoclasts, such as N.Oc, N.Oc/T.Ar, and N.Oc/B.Pm, were consistently reduced (to a significant extent) in CCL5 neuAb-injected bones (Fig. 7f). The parameters of osteoblasts, such as N.Ob, N.Ob/T.Ar, and N.Ob/B.Pm, were also severely reduced by the injection of CCL5 neuAb. These findings demonstrated that the blockade of CCL5 had an impact on the bone metabolism through the inhibition of both osteoclasts and osteoblasts.

## Discussion

The cellular architecture of osteoclasts is unique in that the cytoskeleton of the osteoclast polarizes its resorptive machinery to the bone surface, where an isolated resorptive microenvironment is sealed by a gasket-like structure known as an actin ring or sealing zone. This actin ring is composed of numbers of podosomes, which are dynamic structures that have been implicated in both cellular adhesion and motility. This unique cytoskeletal architecture is regulated by αVβ3 integrin-mediated canonical signaling, which includes the activation of c-Src and its associated molecules such as Pyk2 and Cbl[34–39,47,48], and small GTPases such as Rac and Rho[49–51].

In our in vitro experiments using human and mouse osteoclasts, a loss of the CCR5 function caused abrogated actin ring formation due to the deformation and disassembly of podosomes, which also accompanied the disassociation of the focal adhesion molecules such as Pyk2 and vinculin with podosomes. $Ccr5$-deficient osteoclasts also showed disorganized cellular motility

**Fig. 4** CCL5 stimulates osteoclastogenesis through adhesion signaling. **a** Double immunofluorescence staining of the trabecular bone in the wild-type mouse tibia (4-week-old, male) with anti-Cathepsin K (shown in red) and anti-CCR5 (in green) antibodies. The nuclei were stained with DAPI. Staining of $Ccr5$-deficient bone sections are also shown to confirm the specificity of anti-CCR5 antibodies. Magnification (objective lens) ×40 (left and right panels, wide-field fluorescence microscopy, scale bars, 20 µm) and ×100 (middle panel, SIM, scale bar, 20 µm). BM bone marrow, $n = 3$. **b** The relative mRNA expression levels of $Ccr5$ during osteoclastogenesis were measured by a real-time Q-PCR (mean ± SD, $n = 4$). **c, d** The effect of CCL5 on osteoclastogenesis was revealed by TRAP staining (scale bar, 100 µm). BMCs that were isolated from $Ccr5^{+/+}$ and $Ccr5^{-/-}$ mice were cultured with M-CSF and RANKL with or without recombinant CCL5. The number of multi-nucleated osteoclasts following exogenous CCL5 treatment was quantified (mean ± SD, $n = 4$). *$P < 0.05$ by Student's $t$-test. **e** The effects of CCL5 on focal adhesion signal were investigated by immunoblotting. BMCs isolated from wild-type mice were cultured with M-CSF and RANKL for 3 days and preincubated with CCL5 for 30 min prior to sRANKL stimulation for the indicated time. The phosphorylation levels of FAK, Src, and Pyk2 were analyzed. Immunoblotting data were replicated more than three times. **f** BMCs were treated with RANKL alone or the combination of RANKL and CCL5 (100 ng mL$^{-1}$). Total cell lysates immunoprecipitated with phospho-tyrosine kinase and immunoblotted with phospho-Src. **g** BMCs isolated from wild-type mice were cultured for 3 days with M-CSF and RANKL to induce osteoclast precursors and then treated with rmCCL5 for the indicated time after serum starvation for 30 min. Total cell lysates were harvested to analyze the activity levels of GTP-Rac and -RhoA by a G-LISA (mean ± SD, $n = 5$). *$P < 0.05$ by Student's $t$-test. **h–j** A gene ontology analysis of the genes that showed significantly altered expression levels in RNA-sequencing using cells incubated with RANKL alone and the combination of RANKL and CCL5 (100 ng mL$^{-1}$) (three analyses per condition). The significantly upregulated pathways (**i**) and heat maps of the obtained results (**j**) are shown. Red, high expression; green, low expression

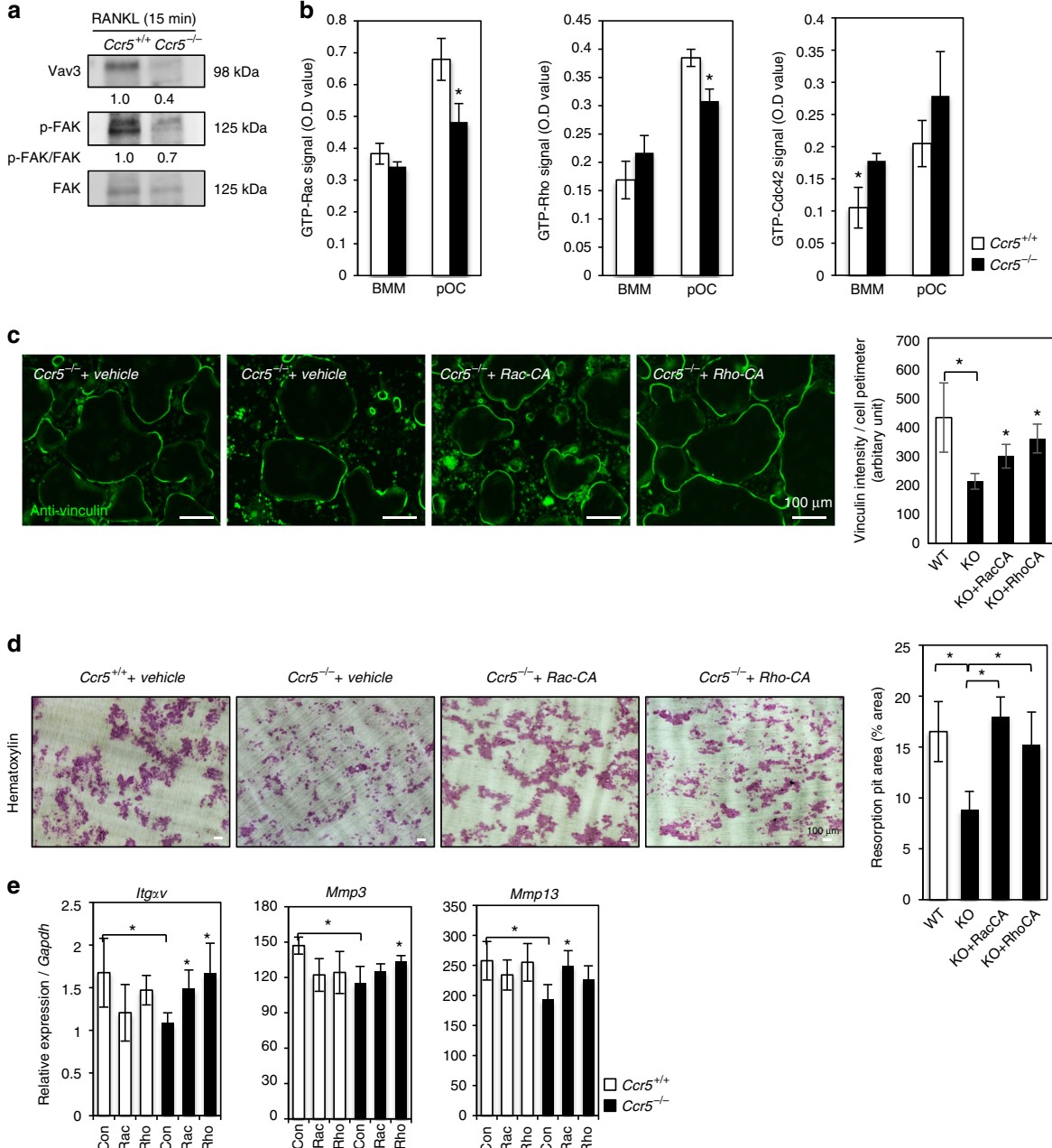

**Fig. 5** The functional rescue of the *Ccr5*⁻/⁻ osteoclasts by active small GTPases. **a** Immunoblotting analyses of Vav3 and phosphorylated FAK were conducted 15 min after RANKL stimulation in wild-type and *Ccr5*⁻/⁻ BMCs. The immunoblotting data were replicated more than three times. **b** The levels of the active forms of Rac1, RhoA, and Cdc42 in osteoclastic differentiation were analyzed in cells from wild-type and *Ccr5*⁻/⁻ bones (mean ± SD, n = 5). **c** The adhesion ring formation of wild-type and *Ccr5*⁻/⁻ osteoclasts expressing the indicated constructs was examined in cells cultured on a glass-bottomed dish, and then analyzed by anti-vinculin immunofluorescence staining (shown in green, scale bars, 100 μm). Magnification ×20 (objective lens), n = 4. The vinculin intensity per cell perimeter was quantified and statistically compared (mean ± SD, n = 10). *P < 0.05 by Student's *t*-test. **d** Resorption pit assays. After culturing, the osteoclasts were removed from the dentin slices and were subsequently stained with hematoxylin to visualize the resorption pits (scale bars, 100 μm, n = 5). The resorption pit areas (%) on images were scored and statistically compared. **e** The relative mRNA levels of *Integrin-αV*, *Mmp3*, and *Mmp13* were measured by a real-time Q-PCR (mean ± SD, n = 4). *P < 0.05 by Student's *t*-test

and a reduced bone resorptive ability, which was further associated with reduction in the RANKL-induced phosphorylation of Src, Pyk2 and FAK, the expression of Vav3, and the GTP forms of Rac and Rho. In contrast, the RANKL-induced phosphorylation of NF-κB and P38 was comparable to that in wild-type cells. These findings suggest that CCR5 selectively sustains the RANKL-induced pathways associated with the integrin-mediated signaling (Supplementary Fig. 9A). Interestingly, the exogenous expression of the constitutively active forms of either Rac or Rho

in *Ccr5*-deficient osteoclasts resulted in a sufficient recovery of bone resorptive activity, suggesting that the dysfunction of *Ccr5*-deficient osteoclasts is mainly due to the disrupted pathway involving these small GTPases, and, in turn, that the activation of these small GTPases is essential for maintaining the spatial assembly of integrin-mediated signaling complex and its activity (Supplementary Fig. 9A).

Chemokine signaling in osteoclastogenesis has not been well documented, although the essential roles of chemokine receptors

in bone metabolism have been reported by our group and others[41,52,53]. CCL5 inoculation stimulated RANKL-induced osteoclast formation through CCR5 in vitro. Exogenous CCL5 also induced the phosphorylation of FAK and the Rac activation. These findings suggest that the activation of CCR5-mediated

signaling stimulates the focal adhesion-mediated signaling. Furthermore, our transcriptome analysis demonstrated that CCL5 transcriptionally increased the numbers of component molecules that generally participate in integrin- and chemokine-mediated signaling, osteoclast differentiation, and function. Taken together,

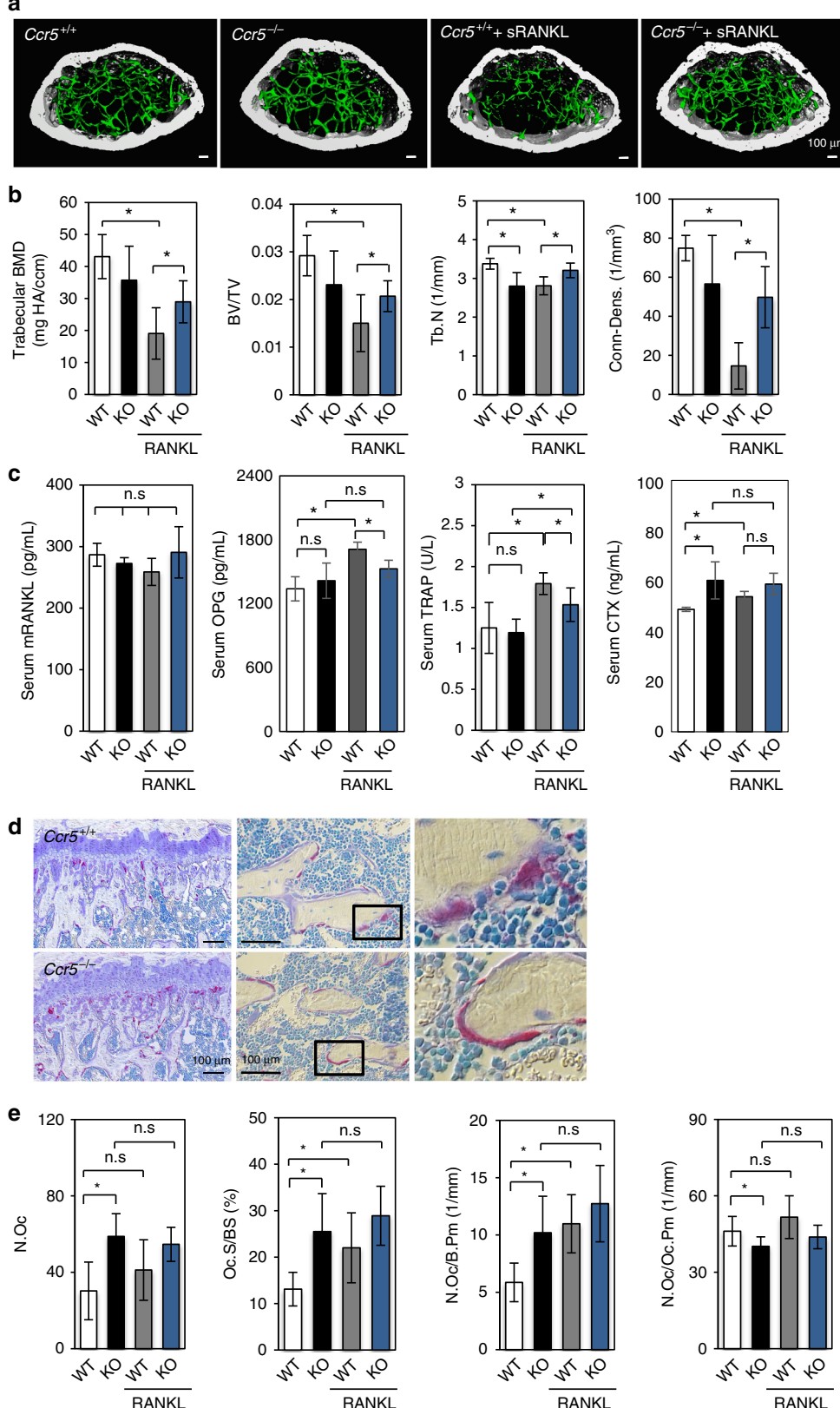

these findings and the above-described results of the loss-of-function experiments suggest that the CCR5-mediated signal activates the focal adhesion-mediated signaling that is required for the functional cytoskeletal architecture of osteoclasts involved in podosome belt organization, cell spreading, cellular locomotion, and bone resorptive activity. This CCR5 signaling is thought to be quite unique to osteoclasts, which is in clear contrast to the previous findings on CCR5 signaling using JAK/STAT and NF-κB in immune cells and cancer cells[54–57].

CCR5 shares the same ligands such as CCL3 and CCL5 with CCR1[1]. Our previous study demonstrated that *Ccr1*-deficiency reduced the number of precursors and impaired the multi-nucleation of osteoclasts[41], indicating that CCR1 is required for the initial differentiation of osteoclasts. In this study, *Ccr5* deficiency did not significantly affect the numbers of osteoclast precursors or the migration activity of BMCs toward M-CSF. *Ccr5*-deficient cells substantially developed multinuclear osteoclasts, but had disorganized podosomes. Temporal transcriptional patterns of *Ccr1* and *Ccr5* during osteoclast differentiation demonstrated the late upregulation of *Ccr5* (Hoshino et al.[41] and the present study). These findings suggest a temporal functional relay of CCR1 and CCR5 during osteoclast differentiation, even though they possibly share the same ligands (Supplementary Fig. 9B).

Recent report demonstrated that Maraviroc was associated with decreased bone loss at the hip and lumbar spine in comparison to patients who received tenofovir disoproxil fumarate (TDF)-containing ART[26]. It is therefore suggested that the pathways in bone cells, rather than the inflammatory and immunomodulatory pathways, may be more involved with the beneficial skeletal effect imparted by the functional loss of CCR5. Accordingly, we first investigated the impact of the functional loss of CCR5 on cultured human osteoclasts and osteoblasts. The blockade of CCR5 by anti-hCCR5 neuAb or the CCR5-antagonist Maraviroc affected the relatively later stages of human osteoclast differentiation associated with impaired actin ring formation and disorganized podosomes, as discussed above. We next took advantage of a model of RANKL-induced osteoporosis, as RANKL is a direct inducer of osteoclastogenesis and thus, directly provides a bone-destructive condition that is independent from inflammatory and immunomodulatory bone loss.

The histomorphometric parameters of osteoclasts indicated that, under regular conditions, increased numbers of osteoclasts compensated for the osteoclast dysfunction in *Ccr5*$^{-/-}$ mice. The cellular proliferation rate of BMCs from *Ccr5*$^{-/-}$ bones in the presence of M-CSF was higher in comparison to that of wild-type cells. Thus, it is possible that the CCR5 deficiency is a condition that is favorable for the expansion of osteoclastic precursor cells prior to RANKL stimulation. Our pathway analyses suggested a possible link between CCR5-mediated signaling and β-catenin. Wei et al.[58] demonstrated the biphasic and dosage-dependent involvement of β-catenin in the proliferation and differentiation of osteoclast precursors. This report together suggests that CCR5 is involved in such switching mechanism in proliferation and differentiation of osteoclast precursors in collaboration with β-catenin.

In our RANKL-induced osteoporosis model, *Ccr5*$^{-/-}$ mice were found to be less susceptible to this osteoporotic stimulation due to the functional loss of their osteoclasts. The histomorphometric parameters showed that bone formation was slightly, but not overtly, stimulated in the *Ccr5*$^{-/-}$ mice under this bone-destructive condition (Supplementary Fig. 8B). These analyses of *Ccr5*$^{-/-}$ bones revealed a bone metabolic condition that was similar to osteoclast-rich osteopetrosis, which is characterized by reduced bone resorption, increased numbers of osteoclast, but normal or even increased bone formation[59].

These findings appear to be contradictory to those of a previous report. In a model of orthodontic tooth movement, the number of TRAP-positive osteoclasts and the expression of osteoclastic markers in *Ccr5*$^{-/-}$ alveolar bone were significantly higher in comparison to the wild-type alveolar bone[30]. The increase in the number of osteoclasts in this model was consistent with our observation. Reports by some of our research group demonstrated that the distinct regulation of alveolar bone remodeling in the tensile and compression sites was triggered by distinct changes in mechanical loading induced by orthodontic force[60,61]. Thus, bone site-specific and functional analyses of osteoclasts would reveal the roles of CCR5 in this model in greater detail.

In pathological bone destruction, several findings reported that CCL3 (also called MIP-1a) that also binds to CCR5 was a major pro-inflammatory chemokine produced at sites of inflammation and stimulated osteoclastogenesis in the context of RA[29,44] and in the bone resorption induced by myeloma[28,45,46]. In these conditions, CCL3 is fundamentally produced by cells other than osteoclasts, thus functions in a paracrine manner. On the other hand, a previous finding demonstrated that CCL5 produced by osteoblasts and osteoclasts have roles in these bone cells in autocrine and paracrine manners[27]. This study observed that the serum CCL5 level was ~60 pg mL$^{-1}$, while the CCL3 level was below the detectable level. Bones from mice injected with a neutralizing antibody against CCL5 at 6 weeks of age showed an increased trabecular bone mass with significantly reduced numbers of osteoclasts and osteoblasts. These data demonstrated that the circulating and extracellular supply of CCL5 was required for bone metabolism.

Wintges et al.[31] reported that the bones of 6-month-old *Ccl5*-deficient mice showed a decreased BV/TV value with impaired bone formation and increased bone resorption. This study reported that the severe decrease in BV/TV was transient as the values measured at 3 and 12 months of age did not differ to a statistically significant extent, while the numbers of osteoclasts remained increased, even at 12 months of age. The decreased number of osteoblasts that was observed after the blockade of CCL5 in this study was possibly consistent with the osteoblastic phenotype in older *Ccl5*-deficent mice, as we often observed osteoblastic cell-free bone surfaces in our experiments, as was observed in *Ccl5*-deficent bone specimens[31]. However, there were

---

**Fig. 6** *Ccr5*-deficient male mice are resistant to RANKL-induced bone loss. **a**, **b** Micro-computed tomography (μCT) images (scale bars, 100 μm) and the analysis of the femurs of 7-week-old *Ccr5*$^{-/-}$ and their wild-type littermates (*Ccr5*$^{+/+}$) (n = 4–6 mice per group). The BMD, BV/TV, Tb.N, and Conn-Dens were scored and statistically compared. **c** The levels of serum mouse RANKL, OPG, TRAP, and CTX were measured by an ELISA, and statistically compared. **d** Representative images of histological sections of the distal femurs obtained from wild-type and *Ccr5*$^{-/-}$ mice are shown. The sections were stained to show the activity level of TRAP (shown in red), and Villanueva staining was performed to reveal the osteoid and osteoblasts (in pale purple); the nuclei were revealed by toluidine blue staining (in blue). The stained sections were observed by differential interference contrast (DIC) microscopy. Magnification (objective lens): ×10 (upper panel), ×20 (middle panel), and ×40 (lower panel), respectively. Scale bars, 100 μm, n = 4–6. **e** Quantitative bone histomorphometric analyses were conducted of the trabecular bones in the distal femurs of *Ccr5*$^{+/+}$ and *Ccr5*$^{-/-}$ mice. The osteoclast number (N. Oc), osteoclast surface per bone surface (Oc.S/BS), osteoclast number per bone perimeter (N.Oc/B.Pm) and osteoclast number per osteoclast perimeter (N.Oc./Oc.Pm.) were scored and statistically compared. Each sample was duplicated. *P < 0.05 (by one-way analysis of variance (ANOVA)) in comparison to WT or RANKL-injected WT. All values are shown as the mean ± SD, n = 4–6

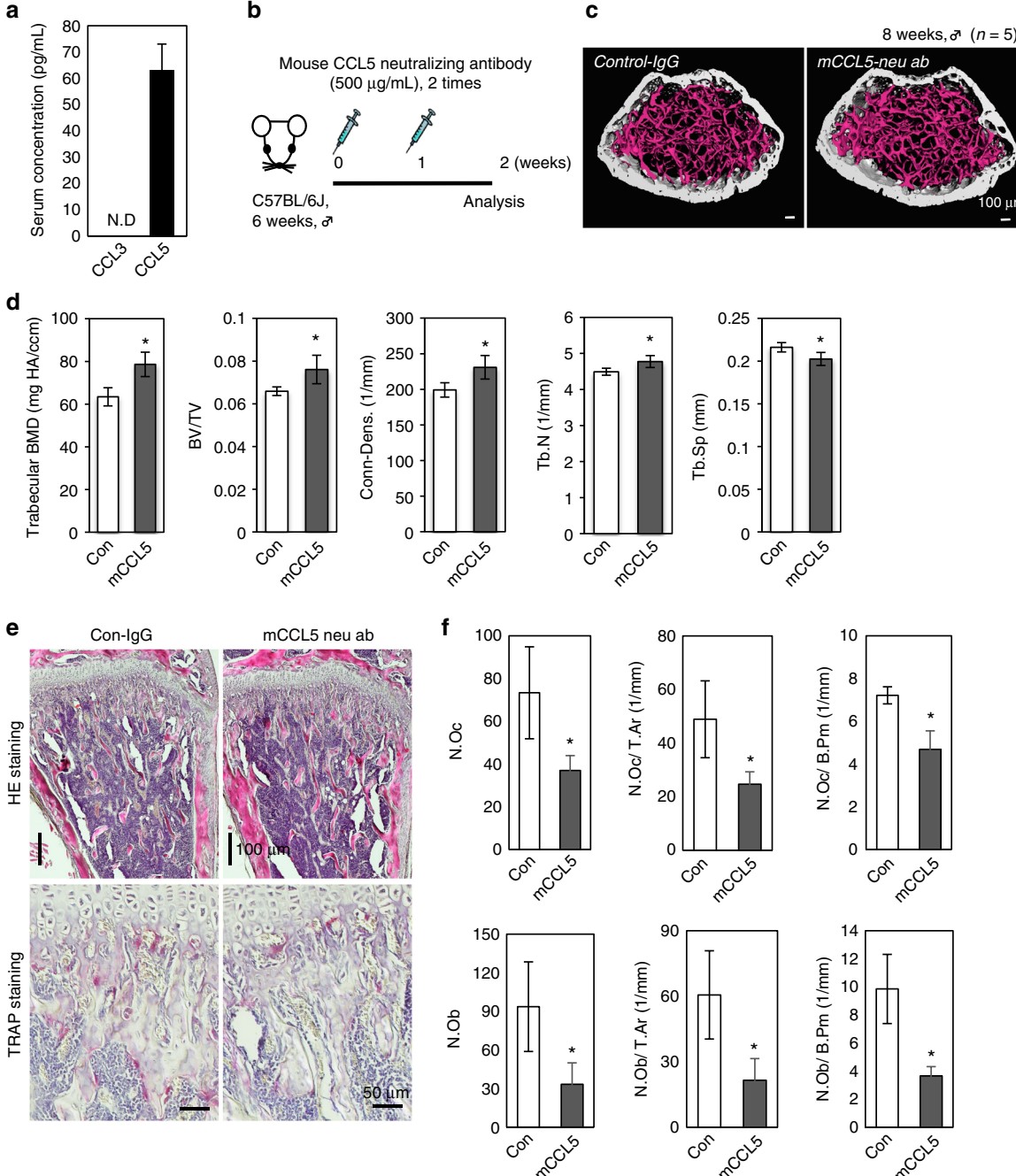

**Fig. 7** The blockade of CCL5 in vivo. **a** The serum levels of CCL3 and CCL5 in 10-week-old mice were measured by an ELISA (*n* = 5). **b** Mouse anti-CCL5 neutralizing antibodies (mCCL5 neu ab, 500 μg per mouse) were injected (once per week for 2 weeks) into 6-week-old male C57BL/6J mice (*n* = 5). IgG was administered to the control group. **c**, **d** μCT images (scale bars, 100 μm) and parameters (*n* = 5 mice per group) are shown. **e** Representative images of the distal femurs from the control IgG and mCCL5 ab groups. HE- (scale bars, 100 μm) and TRAP-stained sections (scale bars, 50 μm) are shown in the upper and lower panels, respectively (*n* = 5). **f** Quantitative bone histomorphometric analyses were conducted of the trabecular bones in the distal femurs of control IgG and mouse CCL5 neutralizing antibody-injected mice. *\**P* < 0.05 (by Student's *t*-test) in comparison to control and mCCL5-neuAb mice. All values are shown as the mean ± SD, *n* = 5

marked differences in the resorption scores in young CCL5-blocked bones and older *Ccl5*-deficient bones. Thus, our blockade of CCL5 in young bones did not fully replicate the phenotype of 6-month-old *Ccl5*-deficient bones, which may be partly explained by age-related functional redundancy as discussed by Wintges et al.[31] The antibody blockade of a chemokine ligand only inhibits *trans*-interaction with its receptors, while the genetic deletion of a ligand can eliminate both *cis*- and *trans*-interaction, which may cause distinct cellular outputs. In the components of G-protein-coupled receptor (GPCR) signaling, the ligand-receptor interaction on the plasma membrane and endosomes, leads to the transduction of distinct signaling and cellular functions[62]. These mechanisms may be involved in the chemokine signaling in bone cells, and might explain why bone phenotype in the CCL5-blocked young mice differed markedly from that in older *CCL5*-deficient mice. Nevertheless, these two experiments revealed the essential role of CCL5 in bone remodeling without an immunological trigger[31]

The severe decrease in the number of osteoclasts following the antibody blockade of CCL5 was similar to the bone phenotype in *Ccr1*-deficient mice[41]. It is possible that some of the same mechanisms are involved in the bone phenotypes observed after the blockade of CCL5 and those observed in *Ccr1*-deficient bones. This might mask the CCR5-dependent phenotypes, as CCR1 has more critical role, especially in the earlier stages of osteoclast differentiation than CCR5 does. Thus, though the blockade of CCL5 did not fully phenocopy osteoclastic dysfunction in *Ccr5*-deficient bones, CCL5 is possibly a relevant ligand for osteoclastogenesis in both CCR1 and CCR5. Taken together, the results of our current study further elucidated the relevance of CCL5 in the bone metabolism.

Conclusively, our findings in this study suggest that CCR5-antagonist treatment may help to substantially improve the quality of life of HIV-infected patients and suggest a new therapeutic approach for preventing various bone-destructive diseases as well as arterial hypertension and anti-tumor immunity by blocking CCR5[63,64].

## Methods

**Animal studies.** All of the animal experiments were performed in accordance with the Institutional Guidelines for the Care and Use of Laboratory Animals in Research and with the approval of the local ethics committees of the University of Tokyo, the Research Institute of National Center for Global Health and Medicine, and Ehime University. Standard 6-week-old male C57BL/6 mice were obtained from CLEA Japan. *Ccr5*-deficient mice (*Ccr5$^{-/-}$*) were generated as previously described[65]. All mice were backcrossed for 8 to 10 generations on the C57BL/6 background mice. The mice were all bred and maintained under pathogen-free conditions at the animal facilities of the University of Tokyo and Ehime University.

**Materials.** Recombinant mouse M-CSF and RANKL were purchased from R&D Systems Inc. (Minneapolis, MN) and Wako Chemicals (Japan), respectively. Recombinant mouse CCL5 (RANTES, 478-MR) and CCL9 (MIP-1γ, 3217-MG), anti-human (MAB182), and anti-mouse (MAB-478) CCL5 neutralizing antibodies were purchased from R&D Systems. For immune staining, anti-CCR5 Ab (ab11466, 1:100), anti-Cathepsin K (ab19027, 1:100), anti-Pyk2 (ab32571, 1:100), anti-vinculin (ab1186, 1:100), anti-CCL5 (ab189841, 1:100), and anti-tubulin (ab6160, 1:100) mAb were purchased from Abcam Inc. (Cambridge, MA). The secondary antibodies  Alexa488 (rabbit A11008, mouse A11029), 568 (rabbit A11011, mouse A11004), and 647 (rabbit A21245, mouse A21236, rat A21247)-labeled IgG (all dilution 1:1000) were purchased from Molecular Probes (Eugene, OR). AlexaFluore 488 (A12379, 1:200) and 568 (A12380, 1:200) phalloidin for actin structure staining were purchased from Invitrogen Corp. Control rabbit IgG (A10042, 1:100), rat IgG (A11006, 1:100), and mouse IgG (A11004, 1:100) were purchased from Molecular Probe. Mouse chemokines were detected using an ELISA with the ab215537 and ab200017 (Abcam Inc.) for CCL5 (RANTES) and CCL3 (MIP-1α). The levels of mouse tartrate-resistant acid phosphatases (TRAP5b) and C-terminal telopeptide and in serum were measured using a mouse TRAP EIA assay kit and CTX-I ELISA kit (Immunodiagnostic system, Fountain Hills, AZ). A Mouse RANKL ELISA kit (R&D system), Mouse Osteocalcin/Bone gla protein, OT/BGP ELISA kit (Biomedical Technologies, Stoughton, MA), M-CSF (R&D system), Osteoprotegerin/TNFRSF11B ELISA kit (R&D system). For immunoblotting, anti-phospho-Src (#6943, 1:1000), anti-Src (#2108, 1:1000), anti-phospho-FAK (#3281, 1:1000), anti-FAK (#3285, 1:1000), anti-phospho-Akt (#9275, 1:1000), anti-Akt (#9272, 1:1000), anti-phosphor-ERK (#9101, 1:1000), anti-ERK (#9102, 1:1000), anti-phospho-p38 (#9211, 1:1000), anti-p38 (#9212, 1:1000), anti-phospho-NF-kB (#8242, 1:1000), anti-NF-kB (#8242, 1:1000), anti-phosphor-Pyk2 (#3291, 1:1000), and anti-Pyk2 (#3292, 1:1000) were purchased form Cell Signaling Technology Inc. (Beverly, MA). Anti-Vav3 (#07-465, 1:1000) were purchased from Millipore. Anti-α-actin (A5316, 1:2000) and anti-GAPDH (#2118, 1:1000) were purchased from Sigma (St. Louis, MO) and Cell Signaling Technology Inc., respectively. The quantitative analyses of Rac and Rho activation were performed using the G-LISA activation assay (Cytoskeleton, Inc., Denver, CO). Maraviroc (S2003) was purchased from Selleck Chemicals (Houston, TX). Control siRNA (Fluorescein Conjugated) and human-siCCR5 (Ambion pre-designed, AM16704) were purchased from Santa Cruz Biotechnology's (Santa Cruz, CA, sc-36869) and Thermo Fisher Scientific (Waltham, MA), respectively. The µ-slides and transwell filters that were used in the chemotaxis experiments were purchased from Ibidi GmbH (Martinsried, Germany) and Corning Costar (Cambridge, MA), respectively. CA-Rac (Rac1 L61) and CA-Rho (RhoA L63) were purchased from Cell Biolabs, Inc. (San Diego, Ca), and Ad5-CMV-GFP was purchased from Applied Viromics (Fremont, CA).

**Cultures of human cells and the blockade of human CCR5.** Normal human natural osteoclast precursor cells (Poietics™ Osteoclast Precursor Cell System, Cat. No. 2T-110) and human mesenchymal stem cells (Poietics™ Human Mesenchymal Stem Cell Systems, Cat. No. PT-2501) were purchased from Lonza Walkersville, Inc. (Walkersville, MD), and were maintained with Osteoclast Precursor Cell Basal Medium (Lonza) in the absence of supplemented growth factors. Multinuclear osteoclasts were cultured according to the manufacturer's instructions; in brief, osteoclasts were cultured with Osteoclast Precursor Cell Basal Medium (Lonza) including both M-CSF and RANKL, supplemented with the indicated concentrations of anti-CCR5 antibodies, and the culture media was replaced every 3 days. Human osteoblastic cells were induced from human mesenchymal stem cells in osteogenic basal medium (Lonza) supplemented with ascorbic acid, β-glycerophosphate, and dexamethasone until harvest on day 14, with replacement once every 3 days in the presence of anti-CCR5 neuAbs.

U-937 cells (ATCC CRL-1593.2), a human lymphocyte cell line, were obtained from American Type Culture Collection (Manassas, VA) and were maintained in RPMI 1640 medium (Gibco BRL, Gaithersburg, MD) supplemented with 10% fetal bovine serum. For differentiating osteoclast-like cells, the cells were plated with RPMI 1640 medium containing 12-*O*-tetradecanoylphobol 12-myristate 13-acetate (TPA) ($10^{-7}$ M) for 24 h and the cell medium was replaced with fresh RPMI 1640 containing $10^{-8}$ M 1,25(OH)$_2$D$_3$ for 48 h[32]. The cellular differentiation of the U-937 cells was monitored by counting the number of TRAP-positive, multinucleated osteoclast-like cells.

**Mouse osteoclast culture.** Mouse bone marrow cells (BMCs) isolated from 4–6-week-old mice cultured in α-MEM (Gibco BRL, Gaithersburg, MD) were used as the source of osteoclasts. Bone marrow-derived macrophages (BMMs) were induced with 50 ng/mL M-CSF and 100 ng/mL RANKL for an additional 5–6 days. The culture media was replaced every 3 days. The TRAP activity in the osteoclasts was determined by staining using TRAP staining kit (Wako). The contamination of stromal/osteoblastic cells was monitored by a Q-PCR, as a low expression level of the *Osteoprotegrin* gene indicates stromal/osteoblastic cells. To assess bone-resorption activity, bone marrow macrophages were loaded onto dentin slices and incubated for 7 days in the presence of M-CSF and RANKL. After the fixation of the cells in paraformaldehyde (pH 7.4), the dentin slices were stained with hematoxylin and Alexa568 conjugated phalloidin for observation.

**Immunocytochemical staining.** For immunocytochemical staining, osteoclasts were cultured in a cover glass chamber, fixed with 4% paraformaldehyde, permeabilized, and stained with the indicated specific Abs or Alexa488-labeled phalloidin (Molecular Probes), followed by Alexa564-conjugated Abs. The osteoclasts with multiple nuclei (>3) were enumerated. Images were captured using an ECLIPSE Ni-E wide-field microscope (Nikon, Tokyo, Japan) and analyzed using the NIS-Elements software program (Nikon). To perform live-cell imaging, osteoclasts were infected with AdV-GFP at an MOI of 50. At 24 h post-infection, time-lapse fluorescence imaging of mature osteoclasts maintained at 37 °C on a heated microscope stage were captured using a Nikon ECLIPSE Ti (Nikon) equipped with a ×20/0.45 extra-long working distance lens and an Andor iXon3 camera (Andor) for 10 h. The images were captured every 10 min using the NIS-Elements (Nikon) and were processed for quantification using the ImageJ software program. Alternatively, super-resolution imaging on fixed samples was performed by SIM, using an N-SIM (Nikon), equipped with a ×100/1.49 TIRF oil immersion objective lens (Nikon) and an Andor iXon+ electron multiplying charged-coupled device camera (Andor Belfast, UK). Image processing was conducted using the NIS-Elements software program (Nikon) and IMARIS (Bitplane, Zurich, Switzerland).

**The real-time Q-PCR.** Total cellular RNA from osteoclasts and osteoblasts was isolated using an RNeasy kit (Qiagen, Valencia, CA). The total RNA was then reverse-transcribed into cDNA using a Superscript III RT kit (Invitrogen, Carlsbad, CA). Real-time quantitative PCR was carried out in an Eco TM Real-Time PCR system (Illumina, SD, USA) using the intercalation dye SYBRGreen I as a fluorescent reporter and a Taqman® Gene expression analysis (Applied Biosystems, Foster City, CA). The sequences were amplified for 40 cycles under two-step conditions: denaturation at 95 °C for 15 s, annealing and extension at 60 °C for 60 s. The gene expression levels were compared to the *Gapdh* gene expression by the $2^{-\Delta(Ct)}$ method. All sequences for human and mouse primers used in the study were provided in Supplementary Tables 1 and 2.

**Protein analysis.** Total cell lysate from osteoclasts was collected using RIPA buffer with a phosphatase inhibitor cocktail and a protease inhibitor cocktail (Sigma). Cell lysates were subjected to SDS/PAGE under reducing conditions and blotted onto a PVDF membrane (Bio-Rad) in semi-dry transfer conditions (Bio-Rad blotting apparatus). After blocking, the membrane was allowed to react with the specific antibodies, and the detection of specific protein was carried out using enhanced chemiluminescence (Amersham Pharmacia Biotech, Buckinghamshire, UK) according to the manufacturer's instructions. The loading differences were normalized using α-actin or GAPDH antibodies. All of the immunoblotting data were

replicated more than three times. The uncropped scans of all Western blots are shown in Supplementary Figs. 10–12.

**The small GTPase assays.** Active GTP forms of Rac1 and RhoA were quantified using a G-LISA™ activation assay kit (Cytoskeleton Inc., Denver, Co.) as specified by the manufacturer. To examine chemokine ligand-induced GTPase activation, BMMs were cultured with M-CSF and RANKL for 4 days. Cells were stimulated with rmCCL5 after serum starvation for 30 min without M-CSF and RANKL.

**Adenovirus-mediated gene transfer.** Primary BMMs were cultured with M-CSF for 24 h and then infected with adenoviral constructs encoding constitutively active form of Rac1 or RhoA at MOI 50 for 36 h. Following transfection and incubation with M-CSF, cells were stimulated with RANKL for next assay.

**Flow cytometry.** Flow cytometry of the osteoclast precursors was performed as previously described[41,52]. Cells were incubated with anti-mouse CD16/CD32 monoclonal antibody (clone 2.4G2; BioXcell, West Lebanon, NH) to block non-specific staining of Fc-receptors, then stained with combinations of fluorophore-conjugated Abs. Abs against mouse CD11b (M1/70), CD24 (M1/69), Ly-6C (HK1.5), CD115 (AFS98), CX3CR1(SA011F11), and CD265/RANK (R12-32) were purchased from BioLegend (San Diego, CA). Data were acquired on a Gallios flow cytometer (Beckman Coulter, Fullerton, CA) and analyzed using FlowJo software (version 10.3; TreeStar, Ashland, OR). Non-viable cells were excluded based on forward and side scatter profiles and propidium iodide staining.

**Chemotaxis migration assays.** 3D chemotaxis assays were performed using μ-Slide Chemotaxis (ibidi, Madison, WI). BMMs were prepared in 1% FBS and cast to attach in the chamber for 30 min before the assays. Images were taken in the incubation chamber every minute for 1.5 h using a Nikon ECLIPSE Ti (Nikon, Tokyo, Japan) equipped with a ×10/0.45 extra-long working distance lens and an Andor iXon3 camera (Andor). Images were imported into the NIS-Elements software program (Nikon). The migration displacement and speed were calculated by cell tracking in the IMARIS software program.

**Osteoporosis induction in mice.** To establish a bone loss model, sRANKL was administered to mice. The littermates of 6-week-old male $Ccr5^{+/+}$ and $Ccr5^{-/-}$ mice ($n = 4$–6) were injected intraperitoneally with sRANKL (2 mg/kg) or PBS (vehicle). After 2 days, the mice were euthanized and blood samples were collected for serum isolation. The femora were fixed with 4% paraformaldehyde for 24 h and reserved in 70% ethanol for the next procedure. To evaluate the effects of anti-mouse CCL5 neuAb on the bone mass, 8-week-old male C57BL/6J mice were prepared ($n = 5$). Anti-mouse CCL5 neuAb and Control rat IgG was intravenously injected once per week for 2 weeks at concentrations of 500 μg per mouse and 250 μg per mouse, respectively.

**Micro-computed tomography and bone histomorphometry.** Micro-computed tomography (μCT) scanning of the proximal tibiae was performed using a μCT-35 (SCANCO Medical AG, Bruittesellen, Switzerland) with a resolution of 6 μm, and the microstructure parameters were three-dimensionally calculated as previously described[66]. The unilateral proximal tibiae were fixed with ethanol and embedded in glycol methacrylate. The blocks were cut into 5-μm-thick sections using a microtome (Leica Microsystems, Wetzlar, Germany). The structural parameters were analyzed at the secondary spongiosa. The sections were stained with Villanueva, TRAP and toluidine blue and were analyzed using a semi-automated system (OsteoMeasure, Decatur, GA). The nomenclature, symbols, and units used in the present study were recommended by the Nomenclature Committee of the American Society for Bone and Mineral Research[67,68].

**RNA-sequencing analysis.** The quality of the extracted RNA was assessed using an Illumina kit (Illumina) according to the manufacturer's instructions. Illumina TruSeq Standard mRNA Sample Prep Kit set A was used for the library preparation. mRNA sequencing was performed using an Illumina MiSeq Reagent kit V3 150 cycle kit with 75 bp paired-end sequencing with a fragment size of ~260 bp, which were trimmed to 75 bp. A Heatmap was obtained using MeV[69] and KEGG pathway analyses were performed DAVID Bioinformatics Resources 6.8[70].

**Statistical analyses.** The data are presented as the mean ± standard deviation (SD) of the indicated number of times an examination was repeated. Statistical significance was determined using the Excel (Microsoft, Redmond, WA, USA) or SPSS (IBM, Armonk, NY, USA) software programs. The data were analyzed using a two-tailed Student's $t$-test or by a one-way analysis of variance (ANOVA). Asterisk indicates significant upregulation or downregulation, respectively ($P < 0.05$) and NS is not significant.

**Data availability.** The RNA-sequencing data of mouse recombinant CCL5 treatment experiment are deposited at Gene Expression Omnibus (GEO, Accession Number: GSE106259). All other data available from the authors upon reasonable request.

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

## Acknowledgements

We express sincere thanks to Kazuaki Tokunaga at Nikon Instech for his microscopy expertise, to the members of ADRES in Ehime University for their experimental support, to Naohito Tokunaga for his technical assistance in performing the RNA-sequencing analysis, and to Noriko Tokai and Miwako Iwai at The Institute of Medical Science, The University of Tokyo (IMSUT) for their expertise on microscopy. This work was supported by Grant-in-Aids for Scientific Research from the Japan Society for the Promotion of Science (JP16K20412 and JP26293392), which were awarded to J.-W.L and T.I., respectively. This work was also partly supported by IMSUT Joint Research Project (2016–2019).

## Author contributions

J.-W.L. designed and conducted most of the experiments and wrote the manuscript; A.H. designed and conducted the initial experiments; T.S. contributed to the image processing and analysis; K.I. provided support for the bone histomorphometry analysis; Sh.U. provided support for the adenovirus-mediated gene transfer experiments; S.U. and K.M. provided *Ccr5⁻/⁻* mice and information for the genotyping analysis and conducted the cell population analyses by flow cytometry; Y.I., Y.K., and Y.A. provided support in evaluating the data; and T.I. supervised the whole project, provided financial support, and wrote the manuscript. All of the authors approved the final manuscript.
