## [Peer Review File · Nature Communications]

Reviewers' comments:

Reviewer #1 (Remarks to the Author):

The study by Lee et al. investigates the role of CCR5, a chemokine receptor and co-receptor for HIV, in bone homeostasis, but in osteoclastogenesis in particular. They show that CCR5 is critical for proper osteoclast function by regulating the actin cytoskeleton through the FAK-Src-Rac axis. Thus, in bone pathologies requiring osteoclast activation, deficiency of CCR5 also results in a rescue of the bone phenotype. Previous studies have already shown a role for CCR5 in osteoclastogenesis. However, this is the first study to show mechanistic insights. Overall, the methods seem well conducted and the figures are nicely done. However, a lot of times, proper quantification and analysis of *in vivo* results are lacking. Thus, even though the topic is of high interest, validation of several aspects of this study using proper statistics is required before final conclusions can be drawn.

Specific comments:

1. Introduction: studies that have previously described effects of CCR5 on osteoclastogenesis and osteoclast-osteoblast-crosstalk are mentioned, but have not been cited. I suggest to at least cite these two papers who already have shown effects of CCR5 on bone and bone cells:

Andrade I Jr1, Taddei SR, Garlet GP, Garlet TP, Teixeira AL, Silva TA, Teixeira MM. CCR5 down-regulates osteoclast function in orthodontic tooth movement. *J Dent Res.* 2009 Nov;88(11):1037-41. doi: 10.1177/0022034509346230.

Wintges K, Beil FT, Albers J, Jeschke A, Schweizer M, Claass B, Tiegs G, Amling M, Schinke T. Impaired bone formation and increased osteoclastogenesis in mice lacking chemokine (C-C motif) ligand 5 (Ccl5). *J Bone Miner Res.* 2013 Oct;28(10):2070-80.

1. Figure 1G: the quantification of the cells with the Ab does not seem to match nicely. Perhaps another alternate picture plus quantification can be found.
2. Was the osteoclast size measured? The authors state that the size of the osteoclasts appeared larger, when this could easily be measured using e.g. ImageJ.
3. Page 11: is there a typo concerning the mouse and human osteoclasts? I assume Sup Fig. 2 refers to the mouse osteoclasts.
4. Page 13: integrin b3 is not reduced in the KO – unlike stated in the text
5. The illustration in Fig. 3D could be omitted. It's a fairly straight forward experiment. Instead, I would suggest quantifying the Western blots. How many replicates were performed? This is not stated in the figure legend. Also the activation of p38 seems reduced, given the increased amount of p38 available and the slightly more faint bands for pp38. Further, it is stated on page 14 that FAK was investigated too, but I cannot find the corresponding Western blot.
6. Figure 4E. The quantification of the number of osteoclasts should be indicated per bone surface (similar to Oc.S/BS). Given the lower BV/TV in the WT+RANKL group, the Oc.N/BS will also increase, which reflects better the serum TRAP levels. High levels of TRAP may indicate high levels of bone resorption. However, CTX should be measured as a true measure of bone resorption, as this displays the collagen-breakdown produces. TRAP is rather an indicator for osteoclast numbers. As CCR5^{-/-} osteoclasts show mostly defective bone resorbing capabilities (at least according to *in vitro* data), it could likely be that osteoclast number is increased *in vivo*, as a compensatory mechanism of decreased osteoclastic bone resorbing activity (similar to conditions like osteoclast-rich osteopetrosis). I think this could be worked on a little more carefully to draw a final conclusion.
7. How do the authors explain that CCR5^{-/-} mice have more osteoclasts, however unchanged TRAP and unchanged BV/TV? What about the osteoblasts?
8. Even though the difference between the osteoclast numbers in KO+RANKL mice is not different from KO+PBS, they are even higher than the WT+RANKL group. Yet, BV/TV is higher in KO+RANKL than in WT+RANKL. Are the osteoblasts the explanation for this discrepancy? Again,

what about CTX? Is it just a matter of inefficient resorption?

9. It seems the other cytokines measured in Suppl Fig 3C are not mentioned in the text. If it is shown (the connection to CCR5 is not clear, however), it should also be mentioned.

10. Figure 5E should be quantified.

11. Figure 5J has a poor resolution and cannot be read. Perhaps a table would be better to list the genes.

12. How does the Wnt pathway fit into the whole story? It came up as a regulated pathway in Fig. 6. However, nothing else is mentioned here. Considering Wnt pathway does also play a role in osteoclastogenesis, this should be discussed.

13. Fig. 7A: again, quantification would be good, as in Fig. 3, it didn't seem like NFkB signaling is involved, however here, it seems lower in CCR5^{-/-} cells.

14. Fig 7C: the rescue of the phenotype is not so clear to me. Please quantify using cell size (the osteoclasts from CCR5^{-/-} + Rho-CA still seem huge; does only Rac-CA fully recover the phenotype?) and podosome integrity.

15. Fig 8D: osteoclast parameters should be quantified and data on osteoblasts should be shown. Is it not surprising that neutralizing CCL5 results in opposite effects than blocking CCR5? If this axis is critical, I would assume similar effects of taking out CCL5 or CCR5. In addition, these data contrast to data published by Wingtes et al. (JBMR 2013), who showed low bone mass in CCL5^{-/-} mice with increased numbers of osteoclasts (would rather fit to the CCR5^{-/-} mice). Also, this discrepancy is not mentioned at all in the discussion, but should be clarified or at least discussed.

16. If CCR1 is also a critical receptor for CCL5, what about CCR1 in osteoclasts? It is also highly expressed in osteoclasts. On page 28 in the discussion, it is only stated that CCL5-CCR5 is a more likely axis for osteoclastogenesis, but what about the CCL5-CCR1 axis?

17. I believe there is no functional proof of autoamplification of CCR5 signaling in osteoclasts presented in this paper. This should be rephrased in the discussion on page 29 or tested experimentally.

18. Page 31: the authors state that high numbers of osteoclasts compensate for dysfunctional CCR5^{-/-} osteoclasts – this should be shown formally using CTX (see comment 8).

19. Discussion page 32: only because CCL3 is not produced by bone cells doesn't mean it cannot act on osteoclasts. Why has this chemokine not been investigated with regards to its signaling through CCR5 in osteoclasts?

Reviewer #2 (Remarks to the Author):

The manuscript by Lee et al studies the relevance for CCR5 in the context of osteoclast function in vitro and in vivo. Authors go a long way to suggest potential mechanisms by which CCR5 contributes to osteoclast function and bone metabolism.

There are several studies already investigating the potential role of CCR5 in the context of osteoclast (and osteoblast) function and bone remodeling. This is alluded to in the introduction (page 5, line 14, but no references given) and in the discussion. This needs to be more fully developed. There are studies suggesting CCR5 function may control positively or negatively osteoclast function in vivo and that CCR1 may be more important for pro-osteoclastogenesis (studies of bone remodeling after infection, orthodontic tooth movement and cancer). Authors should not avoid this discussion.

In Figure 1, authors should show the effects of the antibody on days 2-6 (Figure 1B), but never the TRAP Activity (Figure 1C). I think it is important this is shown to have a feeling how individual treatment times compare to overall treatment. In the same Figure 1C, it seems that at the higher dose tested, the antibody also has an effect on days 0-1 (there is also an apparent drop at 0-2). At least the median/mean values are the same as in days 2-4 and 4-6. I understand there is no statistical difference but there is a lot of variation in these experiments (suggesting that greater n

numbers are needed). This has impact on the subsequent interpretation of results.

Experiments in Figure 7 are really interesting as they suggest that CCR5 is functioning via RAc Rho to mediate its osteoclastogenic effects. Have these genes been transferred to control cells as well? In that case, do they enhance function of CCR5+ cells? Couldn't it be that these genes enhance osteoclast function independently of CCR5 (and can overcome the lack of CCR5)? suggesting parallel pathways. This possibility has not even been discussed.

Overall, I find Figure 6 could be shown as supplementary, as data are not necessary for the overall message of the manuscript.

I do not find Figure 8 to add substantially to the manuscript. It is not well discussed, similar results are not seen in CCR5-deficient mice (authors actually conclude on page 31 that "under regular conditions—increased numbers of osteoclasts compensate for their dysfunction in *Ccr5*^{-/-} mice". Why isn't that seen when antibody treatment is given? I think it would be much more interesting to have seen the effects of the antibody on the RANKL-induced effects *in vivo*. As the authors suggest blockade of CCR5 would provide benefit in patients, this sort of experiment would give a therapeutic side to the manuscript.

Finally, the discussion should really encompass other studies on CCR5 and osteoclastogenesis showing the many new findings from the current study. Several studies do suggest that CCR1 may be very relevant for bone remodeling *in vivo*. This should be better discussed.

Reviewer #3 (Remarks to the Author):

In this manuscript the authors report the relevance of CCL5:CCR5 axis in the osteoclasts and show that blockade of CCR5 in human OC cultures or use of CCR5 null mice disrupts actin dynamics and reduces bone resorption. By contrast CCL5 increases OC numbers by leading to expression of signaling pathways involved in cell differentiation, adhesion, ECM-interaction and lysosome secretion. Importantly the *in vitro* studies are supported by *in vivo* findings. However, additional experiments should be performed to confirm the mechanistic and functional studies. Furthermore, the title implicates that CCR5 activation is involved in bone loss in HIV patients, however the authors do not present any data in support of this assumption in the context of HIV infection.

Fig1. Is the effect of the neutralizing Ab reversible? What happens when the media containing a-CCR5 is removed and replaced with normal medium? Do they form functional osteoclasts?

The specificity of the antibody needs to be demonstrated by analyzing the effects of the neutralizing Ab in human cells lacking CCR5. This is particularly important as it seems that there are differences between the morphology of CCR5^{-/-} murine OCs and anti-CCR5 treated hOCs.

OC markers and RANKL-induced signaling should be evaluated in the cells treated with a-CCR5 as it is not clear whether osteoclastogenesis is affected or just the cell spreading.

Podosomes are highly dynamic actin structures that can be rapidly assembled and disassembled. To test whether a-CCR5 has direct effects on podosome dynamics, the authors should add the neutralizing Ab to cultures of day6 OCs and study the actin dynamics then. Furthermore, they should also examine how a-CCR5 affects actin ring reassembly after exposing WT mature OCs plated on bone to cold PBS and allowing the actin rings to reform when warm medium + RANKL and M-CSF is added back.

Fig. 2 Quantification of actin rings (size and/or numbers) should be included.

Previous findings related to defects in podosome organizations showed aberrant formation of the avb3/src/cbl/pyk2 complex. Is the assembly and activation of this complex normal in CCR5^{-/-} OCs?

Fig3. In 3A the authors analyzed steady-state movements of WT and CCR5^{-/-}. Usually OCs respond to chemotactic stimuli such as M-CSF, which does not bind to CCR5, and form lamellipodia. Authors should examine actin dynamics and perform migration assays in response to M-CSF.

Protein levels of integrin subunits need to be evaluated by WB as reduced av transcripts might not necessarily lead to decreased avb3 levels.

Signaling experiments in E should be repeated with M-CSF stimulation and adhesion onto avb3 ligands, which are stronger modulators of actin dynamics than RANKL. Authors mentioned in the text that they measured FAK activation (but it is not shown). However in OCs Pyk2 is most highly expressed rather than FAK, thus they should include Pyk2 in their analysis.

Fig4. The difference in trabecular number between WT and Ccr5^{-/-} in vehicle treated mice is not very convincing especially since no other effects on bone parameters are noted.

What are the effects of CCR5 deletion on bone formation? Also, is the increased in OC number do to higher RANKL and MCSF levels and/or reduced OPG? Does CCR5 deletion affect the number of OC precursors in the bone marrow?

Fig 5. The authors show that CCL5, through binding to CCR5, enhances OCgenesis and RANKL-induced FAK and Src activation. The authors should measure Pyk2 which is the main kinase involved in the OC adhesion complex together with c-Src. Also they should analyze whether CCL5 modulates podosome assembly and actin dynamics as shown in earlier figures. The effects of CCL5 stimulation on bone resorption in mature OCs should also be examined.

Genes in 5J are illegible. Authors also needs to specify what cells were used (BMM, preOCs or mature OV) and for how long they were treated with CCL5

Fig 6 should be included in Sup figures as the authors do not prove that CCR5 is affecting the integrin complex and the signaling networks reported in the figure.

Fig7. Rac and Rho modulate lamellipodia and membrane ruffles. The fig shown in 7C where the authors indicate rescue of the actin ring is too small to appreciate differences between the various groups. For example it is difficult to appreciate any difference between WT and KO + or minus Rac-CA.

Do active Rho and Rac also affect WT cells in terms of integrin expression?

Fig 8. In vivo data need to be expanded by adding OC count, and including OC morphology. What are the effects of CCL5 on OB functions? What is the source of CCL5 in vivo and what are the levels of CCL5 in mice?

Number of mice for in vivo experiments need to be reported.

Response to Referees' comments:

Reviewer #1 (Remarks to the Author):

The study by Lee et al. investigates the role of CCR5, a chemokine receptor and co-receptor for HIV, in bone homeostasis, but in osteoclastogenesis in particular. They show that CCR5 is critical for proper osteoclast function by regulating the actin cytoskeleton through the FAK-Src-Rac axis. Thus, in bone pathologies requiring osteoclast activation, deficiency of CCR5 also results in a rescue of the bone phenotype. Previous studies have already shown a role for CCR5 in osteoclastogenesis. However, this is the first study to show mechanistic insights. Overall, the methods seem well conducted and the figures are nicely done. However, a lot of times, proper quantification and analysis of in vivo results are lacking. Thus, even though the topic is of high interest, validation of several aspects of this study using proper statistics is required before final conclusions can be drawn.

We wish to express our thanks to Reviewer #1 for the constructive comments. Our answers to the specific comments are shown below.

Specific comments:

1. Introduction: studies that have previously described effects of CCR5 on osteoclastogenesis and osteoclast-osteoblast-crosstalk are mentioned, but have not been cited. I suggest to at least cite these two papers who already have shown effects of CCR5 on bone and bone cells:

Andrade I Jr1, Taddei SR, Garlet GP, Garlet TP, Teixeira AL, Silva TA, Teixeira MM. CCR5 down-regulates osteoclast function in orthodontic tooth movement. J Dent Res. 2009 Nov;88(11):1037-41. doi: 10.1177/0022034509346230.

Wintges K, Beil FT, Albers J, Jeschke A, Schweizer M, Claass B, Tiegs G, Amling M, Schinke T. Impaired bone formation and increased osteoclastogenesis in mice lacking chemokine (C-C motif) ligand 5 (Ccl5). J Bone Miner Res. 2013 Oct;28(10):2070-80.

We have introduced these works in the Introduction of the revised manuscript, and have substantially discussed these papers and our results in the Discussion.

1. Figure 1G: the quantification of the cells with the Ab does not seem to match nicely. Perhaps another alternate picture plus quantification can be found.

We have replaced the images in Figure 1G. We have also re-quantified our data with the area, rather than the surface area.

2. Was the osteoclast size measured? The authors state that the size of the osteoclasts appeared larger, when this could easily be measured using e.g. ImageJ.

We measured and compared the size of osteoclasts from wild-type and Ccr5-deficient mice. These data now appear in Sup. Figure 3.

3. Page 11: is there a typo concerning the mouse and human osteoclasts? I assume Sup Fig. 2 refers to the mouse osteoclasts.

This was a typo. This has been deleted in the revised manuscript.

4. Page 13: integrin b3 is not reduced in the KO – unlike stated in the text

As indicated, there was a discrepancy in our previous manuscript with regard to the data that were presented on this point and the description. In the previous version of our manuscript, the data shown in Figure 3C were obtained on Day 5. We have added the data obtained on Day 4 in the revised manuscript. In fact, the integrin b3 level was reduced on Day 4. Accordingly, the statement in the text has been changed.

5. The illustration in Fig. 3D could be omitted. It's a fairly straight forward experiment. Instead, I would suggest quantifying the Western blots. How many replicates were performed? This is not stated in the figure legend. Also the activation of p38 seems reduced, given the increased amount of p38 available and the slightly more faint bands for pp38. Further, it is stated on page 14 that FAK was investigated too, but I cannot find the corresponding Western blot.

As suggested, we have removed the illustration from Figure 3. We performed the quantification using Western blotting. Western blotting was replicated more than three times in each experiment; this is now stated in the Figure legends. We did not investigate FAK in this section; thus, this statement was removed from the text.

6. Figure 4E. The quantification of the number of osteoclasts should be indicated per bone surface (similar to Oc.S/BS). Given the lower BV/TV in the WT+RANKL group, the Oc.N/BS will also increase, which reflects better the serum TRAP levels. High levels of TRAP may indicate high levels of bone resorption. However, CTX should be measured as a true measure of bone resorption, as this displays the collagen-breakdown produces. TRAP is rather an indicator for osteoclast numbers. As CCR5^{-/-} osteoclasts show mostly defective bone resorbing capabilities (at least according to in vitro data), it could likely be that osteoclast number is increased in vivo, as a compensatory mechanism of decreased osteoclastic bone resorbing activity (similar to conditions like osteoclast-rich osteopetrosis). I think this could be worked on a little more carefully to draw a final conclusion.

We totally agreed with comments. The N.Oc/BS value would provide a better insight to our study. However, unfortunately, we could not figure out how to quantify the number of osteoclasts per bone surface (N.Oc/BS). Our semi-automatic morphometric system cannot retrieve No.Ob/BS, probably because N.Ob is a two-dimensional parameter while BS is a three-dimensional parameter. We really apologize for this. We would need more time to achieve this.

We measured the CTX levels, which now appears in Figure 4C. In our data, the CTX levels in KO + RANKL and WT + RANKL were comparable, which may not reflect their BV/TV. We have added the osteoblast parameters to Sup. Figure 4B. The OS/BS and Ob.S/BS values in KO + RANKL were higher than those in KO + vehicle. The osteoclast-rich osteoporosis-like condition is addressed in the Discussion.

7. How do the authors explain that CCR5^{-/-} mice have more osteoclasts, however unchanged TRAP and unchanged BV/TV? What about the osteoblasts?

The enhanced proliferation of CCR5^{-/-} bone marrow cells cultured with M-CSF that we observed may explain the increased number of osteoclasts in CCR5^{-/-} mice. Our pathway analysis suggested a possible link between CCR5 signaling and Wnt / β -Catenin pathway that is reportedly involved in the proliferation-differentiation switching of osteoclasts. These are mentioned in the Discussion. As we noted in Comment 6, the histomorphometric osteoblast parameters are shown in Sup. Figure 4B.

8. Even though the difference between the osteoclast numbers in KO+RANKL mice is not different from KO+PBS, they are even higher than the WT+RANKL group. Yet, BV/TV is higher in KO+RANKL than in WT+RANKL. Are the osteoblasts the explanation for this discrepancy? Again, what about CTX? Is it just a matter of inefficient resorption?

As we answered in comments 6 and 7, the CTX data and the histomorphometric data related to the osteoblasts are shown in Figure 4C and Sup. Figure 4B, respectively. The osteoblast parameters did not show significant differences between WT and KO. Our conclusion is that this can be due to inefficient resorption of CCR5^{-/-} osteoclasts.

9. It seems the other cytokines measured in Suppl Fig 3C are not mentioned in the text. If it is shown (the connection to CCR5 is not clear, however), it should also be mentioned.

We removed the transcriptional levels of human chemokines and chemokine receptors. In the revised manuscript, Sup. Figure 5 only shows the mouse data. This is stated and discussed in the revised manuscript.

10. Figure 5E should be quantified.

The Western blotting data shown in Figure 5E are now quantified by the signal intensity.

11. Figure 5J has a poor resolution and cannot be read. Perhaps a table would be better to list the genes.

The resolution of Figure 5J has been increased.

12. How does the Wnt pathway fit into the whole story? It came up as a regulated pathway in Fig. 6. However, nothing else is mentioned here. Considering Wnt pathway does also play a role in osteoclastogenesis, this should be discussed.

The Wnt / β -Catenin pathway in osteoclastogenesis and the possible association with the CCR5 pathway is discussed in the revised manuscript, as answered to the comment 7.

13. Fig. 7A: again, quantification would be good, as in Fig. 3, it didn't seem like NF κ B signaling is involved, however here, it seems lower in CCR5^{-/-} cells.

The Western blotting data that were shown in Figure 7A in the previous version of our manuscript have been quantified; these data are shown in Figure 6A of the revised manuscript. The NFkB data were removed due to a significant change.

14. Fig 7C: the rescue of the phenotype is not so clear to me. Please quantify using cell size (the osteoclasts from CCR5^{-/-} + Rho-CA still seem huge; does only Rac-CA fully recover the phenotype?) and podosome integrity.

The data that were shown in Figure 7C in the previous version of our manuscript have been quantified and are shown in Figure 6C.

15. Fig 8D: osteoclast parameters should be quantified and data on osteoblasts should be shown. Is it not surprising that neutralizing CCL5 results in opposite effects than blocking CCR5? If this axis is critical, I would assume similar effects of taking out CCL5 or CCR5. In addition, these data contrast to data published by Wingtes et al. (JBMR 2013), who showed low bone mass in CCL5^{-/-} mice with increased numbers of osteoclasts (would rather fit to the CCR5^{-/-} mice). Also, this discrepancy is not mentioned at all in the discussion, but should be clarified or at least discussed.

In the revised manuscript, we have added the osteoclast and osteoblast parameters. As indicated, there were some discrepancies between these data, our CCR5^{-/-} bone data and the CCL5^{-/-} bone data reported by Wingtes et al. These issues are now substantially discussed in the revised manuscript.

16. If CCR1 is also a critical receptor for CCL5, what about CCR1 in osteoclasts? It is also highly expressed in osteoclasts. On page 28 in the discussion, it is only stated that CCL5-CCR5 is a more likely axis for osteoclastogenesis, but what about the CCL5-CCR1 axis?

We previously reported the critical roles of CCR5 in osteoclastogenesis and bone metabolism (Hoshino et al. JBC 2010). The possible relationships between CCR1, CCR5 and CCL5 in osteoclastogenesis are fully discussed in the revised manuscript.

17. I believe there is no functional proof of autoamplification of CCR5 signaling in osteoclasts presented in this paper. This should be rephrased in the discussion on page 29 or tested experimentally.

This description has been changed to “ensuring”.

18. Page 31: the authors state that high numbers of osteoclasts compensate for dysfunctional CCR5^{-/-} osteoclasts – this should be shown formally using CTX (see comment 8).

This is related to Comment 8. We have added the CTX data and reevaluated the phenotype.

19. Discussion page 32: only because CCL3 is not produced by bone cells doesn't mean it cannot act on osteoclasts. Why has this chemokine not been investigated with regards to its signaling through CCR5 in osteoclasts?

We agreed with this point. We have measured the serum levels of CCL3 and CCL5, which are shown in revised Figure 7A. The CCL5 concentration was approximately 60 pg/mL, but the CCL3 concentration was below the limit of detection. This issue is also fully discussed in the revised manuscript.

Reviewer #2 (Remarks to the Author):

The manuscript by Lee et al studies the relevance for CCR5 in the context of osteoclast function in vitro and in vivo. Authors go a long way to suggest potential mechanisms by which CCR5 contributes to osteoclast function and bone metabolism.

We wish to express our thanks to Reviewer #2 for the constructive comments. Our answers to specific comments are below.

There are several studies already investigating the potential role of CCR5 in the context of osteoclast (and osteoblast) function and bone remodeling. This is alluded to in the introduction (page 5, line 14, but no references given) and in the discussion. This needs to be more fully developed. There are studies suggesting CCR5 function may control positively or negatively osteoclast function in vivo and that CCR1 may be more important for pro-osteoclastogenesis (studies of bone remodeling after infection, orthodontic tooth movement and cancer). Authors should not avoid this discussion.

We have given references to the indicated description in the Introduction. We have cited the two following papers in the revised manuscript:

Andrade et al. CCR5 down-regulates osteoclast function in orthodontic tooth movement. J Dent Res. 2009,

Wintges et al.. Impaired bone formation and increased osteoclastogenesis in mice lacking chemokine (C-C motif) ligand 5 (Ccl5). J Bone Miner Res. 2013 Oct;28(10):2070-80.

We have introduced these works in the Introduction of the revised manuscript, and have substantially discussed these papers and our results in the Discussion.

The functional differences between CCR1 and CCR5 (even though they share the same ligands) are addressed in the Discussion.

In Figure 1, authors should show the effects of the antibody on days 2-6 (Figure 1B), but never the TRAP Activity (Figure 1C). I think it is important this is shown to have a feeling how individual treatment times compare to overall treatment. In the same Figure 1C, it seems that at the higher dose tested, the antibody also has an effect on days 0-1 (there is also an apparent drop at 0-2). At least the median/mean values are the same as in days 2-4 and 4-6. I understand there is no statistical difference but there is a lot of variation in these experiments (suggesting that greater n numbers are needed). This has impact on the subsequent interpretation of results.

Thank you very much for this comment. We understand that these data have an impact on the subsequent interpretation of the results of this work. Accordingly, we have repeated this experiment, with a statistical comparison of the TRAP activity of D2-6. The final results are now shown in Figure 1B-C. The data now more clearly demonstrate that the blockade of CCR5 affects relatively later stages of human osteoclast differentiation. Furthermore, we have examined the effects of anti-CCR5 antibodies on cellular proliferation, and observed that high-dose (20 $\mu\text{g}/\text{mL}$) anti-CCR5 antibody treatment had some toxic effects; 1 $\mu\text{g}/\text{mL}$ and 10 $\mu\text{g}/\text{mL}$, as suggested by the manufacturers protocol, were confirmed to be appropriate doses. The specific effects of CCR5 antibodies were also confirmed using CCR5-knocked-down cells. These data are now shown in Sup. Figure 1A and B.

Experiments in Figure 7 are really interesting as they suggest that CCR5 is functioning via Rac Rho to mediate its osteoclastogenic effects. Have these genes been transferred to control cells as well? in that case, do they enhance function of CCR5+ cells? Couldn't it be that these genes enhance osteoclast function independently of CCR5 (and can overcome the lack of CCR5)? suggesting parallel pathways. This possibility has not even been discussed.

This comment is very important for the proper interpretation of our data in Figure 7 (in the previous version of our manuscript). We have added the data for control wild-type cells, which are now shown in Figure 6. The previous Figure 6 was moved to Supplementary Data, in accordance with the suggestion in the next comment. In fact, the upregulation of *Itg β 3* by CA-Rh and CA-Rac was observed in both wild-type and *Ccr5*^{-/-} cells, indicating that the function was independent from CCR5. However, the effects of these constructs on other markers, such as *Itg α v*, *Mmp3* and *Mmp13*, were specific to *Ccr5*^{-/-} cells, suggesting CCR5-dependent effects. These results are fully described and discussed in the revised manuscript.

Overall, I find Figure 6 could be shown as supplementary, as data are not necessary for the overall message of the manuscript.

The data of the pathway analyses shown in Figure 6 in the previous version of our manuscript have been moved to Sup. Figure 8.

I do not find Figure 8 to add substantially to the manuscript. It is not well discussed, similar results are not seen in CCR5-deficient mice (authors actually conclude on page 31 that "under regular conditions—increased numbers of osteoclasts compensate for their dysfunction in *Ccr5*^{-/-} mice". Why isn't that seen when antibody treatment is given? I think

it would be much more interesting to have seen the effects of the antibody on the RANKL-induced effects in vivo. As the authors suggest blockade of CCR5 would provide benefit in patients, this sort of experiment would give a therapeutic side to the manuscript.

We added the morphometric data from the experiments. These data are shown in Figure 7D and F. The phenotypic discrepancies among the anti-CCL5-injected bones, the *Ccl5*^{-/-} bones reported by Wintges et al. in JBMR 2013, and the *Ccr5*^{-/-} bone in the present study are fully addressed in the Discussion section.

In terms of the beneficial effects of the blockade of CCR5, we have tested the effects of Maraviroc on human osteoclasts. These data are shown in Sup. Figure 1C. The benefits obtained in patients by the blockade of CCR5 are further addressed in the Discussion of the revised manuscript.

Finally, the discussion should really encompass other studies on CCR5 and osteoclastogenesis showing the many new findings from the current study. Several studies do suggest that CCR1 may be very relevant for bone remodeling in vivo. This should better discussed.

We have cited additional references and discussed our data. The relationship between the roles of CCR1 and CCR5 in bone remodeling is also described in greater detail in the Discussion.

Reviewer #3 (Remarks to the Author):

In this manuscript the authors report the relevance of CCL5:CCR5 axis in the osteoclasts and show that blockade of CCR5 in human OC cultures or use of CCR5 null mice disrupts actin dynamics and reduces bone resorption. By contrast CCL5 increases OC numbers by leading to expression of signaling pathways involved in cell differentiation, adhesion, ECM-interaction and lysosome secretion. Importantly the *in vitro* studies are supported by *in vivo* findings. However, additional experiments should be performed to confirm the mechanistic and functional studies. Furthermore, the title implicates that CCR5 activation is involved in bone loss in HIV patients, however the authors do not present any data in support of this assumption in the context of HIV infection.

We wish to express our thanks to Reviewer #3 for the constructive comments. In terms of the title, the implication of the association between the loss of CCR5 and resistance to bone loss does not necessarily mean that CCR5 activation is involved in bone loss in HIV patients. However, our data suggest that CCR5 activation is more generally involved in the osteoclast function, and thus bone loss *in vivo*. This background is described in the Introduction. The possible merits of the blockade of CCR5 in HIV patients are further addressed in the Discussion with reference to our data and the findings of previous reports. We have conducted the experiments and analyses that were suggested by Reviewer #3. Our answers to the specific comments are shown below.

Fig1. Is the effect of the neutralizing Ab reversible? What happens when the media containing a-CCR5 is removed and replaced with normal medium? Do they form functional osteoclasts?

We tested whether the effect of the neutralizing Ab was reversible by investigating the effects of ice-cold PBS shock on human osteoclasts, as was suggested in the following comment. In fact, the effect of the neutralizing Ab was reversible. These data are now shown in Sup. Figure 1D and E of the revised manuscript.

The specificity of the antibody needs to be demonstrated by analyzing the effects of the neutralizing Ab in human cells lacking CCR5. This is particularly important as it seems that there are differences between the morphology of CCR5^{-/-} murine OCs and anti-CCR5 treated hOCs.

We confirmed the specificity of the antibody with CCR5-knocked-down human osteoclasts (knock-down was achieved using a specific siRNA). The monitoring of TRAP activity revealed no additional effects of CCR5 antibodies in CCR5-knocked-

down cells, thus confirming the specificity of the antibody. These data are shown in Sup. Figure 1B.

OC markers and RANKL-induced signaling should be evaluated in the cells treated with a-CCR5 as it is not clear whether osteoclastogenesis is affected or just the cell spreading.

OC markers, such as *CathepsinK*, *NFAT-C1* and *TRAP*, were evaluated, and the results are shown in Sup. Figure 1C. In fact, the blockade of CCR5 affects the expression of these markers and the podosome dynamics.

Podosomes are highly dynamic actin structures that can be rapidly assembled and disassembled. To test whether a-CCR5 has direct effects on podosome dynamics, the authors should add the neutralizing Ab to cultures of day6 OCs and study the actin dynamics then. Furthermore, they should also examine how a-CCR5 affects actin ring reassembly after exposing WT mature OCs plated on bone to cold PBS and allowing the actin rings to reform when warm medium + RANKL and M-CSF is added back.

We conducted the experiment that the reviewer suggested. The data are shown in Sup. Figure 1D and E. In fact, normal differentiation medium with RANKL and M-CSF restored the actin dynamics of the cells—which were lost after anti-CCR5 antibody treatment.

Fig. 2 Quantification of actin rings (size and/or numbers) should be included.

The actin rings on dentin slices were quantified. The results are shown in Figure 2B of the revised manuscript.

Previous findings related to defects in podosome organizations showed aberrant formation of the *avb3/src/cbl/pyk2* complex. Is the assembly and activation of this complex normal in CCR5^{-/-} OCs?

The activation of Pyk2 in wild-type and *Ccr5*^{-/-} cells was compared. The results are shown in Figure 3E. The assembly of Pyk2 in podosome belts were analyzed in wild-type and *Ccr5*^{-/-} cells by immunofluorescence and SIM-based microscopy. The data are shown in Figure 3G.

Fig3. In 3A the authors analyzed steady-state movements of WT and CCR5^{-/-}. Usually OCs respond to chemotactic stimuli such as M-CSF, which does not bind to CCR5, and form lamellipodia. Authors should examine actin dynamics and perform migration assays in response to M-CSF.

The migration activities of wild-type and Ccr5^{-/-} cells in response to M-CSF were analyzed by a live imaging assay and a trans-well filter assay. We did not observe a significant difference between wild-type and Ccr5^{-/-} cells in these assays. These data are shown in Sup. Figure 6C and D.

Protein levels of integrin subunits need to be evaluated by WB as reduced av transcripts might not necessarily lead to decreased avb3 levels.

The protein levels of integrins αv and $\beta 3$ were evaluated. The results are shown in Figure 3D.

Signaling experiments in E should be repeated with M-CSF stimulation and adhesion onto avb3 ligands, which are stronger modulators of actin dynamics than RANKL. Authors mentioned in the text that they measured FAK activation (but it is not shown). However in OCs Pyk2 is most highly expressed rather than FAK, thus they should include Pyk2 in their analysis.

As suggested, the phosphorylation of Pyk2 was analyzed with osteopontin (a integrin ligand). These data are shown in Figure 3F.

Fig4. The difference in trabecular number between WT and CCr5^{-/-} in vehicle treated mice is not very convincing especially since no other effects on bone parameters are noted.

As indicated, Tb.N was the only parameter that showed a significant reduction in Ccr5^{-/-} vehicle-treated mice in comparison to WT mice. BMD, BV/TV and Conn.Dens. showed similar tendencies, albeit no significant differences. We think these data are related to the increases in the N. Oc, Oc.S/BS and N.Oc/B.pm values, since osteoblast parameters such as OS/BS, O.th, Ob.S/BS, N.Ob, N.Ob/T.Ar, N.Ob/B.Pm and N.Ob/Ob.Pm did not differ to a statistically significant extent between vehicle-injected WT and CCr5^{-/-} mice. These osteoblast parameters are shown in Sup. Figure 4B.

What are the effects of CCR5 deletion on bone formation? Also, is the increased in OC number do to higher RANKL and MCSF levels and/or reduced OPG? Does CCR5 deletion affect the number of OC precursors in the bone marrow?

As answered above, the osteoblast parameters are shown in Sup. Figure 4B. No significant differences were observed between vehicle-injected WT and Ccr5^{-/-} mice. The serum levels of RANKL, OPG and MCSF were analyzed and are shown in Figure 4C and Sup. Figure 4B. These parameters did not differ to a statistically significant extent between vehicle-injected WT and Ccr5^{-/-} mice.

The differentiation potential of the OC precursors in the bone marrow was analyzed by a FACS-based analysis. The results are shown in Sup. Figure 6A and B. The differentiation potential of the wild-type and Ccr5^{-/-} cells did not differ to a statistically significant extent. However, the cellular proliferation in response to M-CSF was higher in Ccr5^{-/-} cells than it was in WT cells. These results are shown in Sup. Figure 3A.

Fig 5. The authors show that CCL5, through binding to CCR5, enhances OCgenesis and RANKL-induced FAK and Src activation. The authors should measure Pyk2 which is the main kinase involved in the OC adhesion complex together with c-Src. Also they should analyze whether CCL5 modulates podosome assembly and actin dynamics as shown in earlier figures. The effects of CCL5 stimulation on bone resorption in mature OCs should also be examined.

As suggested, we analyzed the phosphorylation of Pyk2 by CCL5. These data are shown in Figure 5E in the revised manuscript. In fact, the temporal dynamics of Pyk2 after CCL5 stimulation were similar to the temporal dynamics of Src. CCL5 stimulated actin ring formation and the resorption of mature osteoclasts on dentine slices. These results are shown in Sup. Figure 7A.

Genes in 5J are illegible. Authors also needs to specify what cells were used (BMM, preOCs or mature OV) and for how long they were treated with CCL5

We replaced the images in Figure 5J with higher resolution images. In this analysis, preOCs were incubated with or without CCL5 for 24h. We specified the cell type and CCL5 treatment in the Results section and the corresponding Figure legend.

Fig 6 should be included in Sup figures as the authors do not prove that CCR5 is affecting the integrin complex and the signaling networks reported in the figure.

The schematic illustration of the pathway analyses in Figure 6 of the previous version has been moved to Sup. Figure 8 in the revised manuscript.

Fig7. Rac and Rho modulate lamellipodia and membrane ruffles. The fig shown in 7C where the authors indicate rescue of the actin ring is too small to appreciate differences between the various groups. For example it is difficult to appreciate any difference between WT and KO + or minus Rac-CA.

Do active Rho and Rac also affect WT cells in terms of integrin expression?

We have added the control wild-type cell data. These are now shown in Figure 6. Adhesion ring formation, which was analyzed by anti-Vinculin staining, was statistically compared. These data are shown in Figure 6C.

The transcriptional levels of integrins and Mmps in wild-type cells expressing Rac-CA or Rho-CA have been analyzed. In fact, the upregulation of *Itgβ3* was observed by CA-Rh and CA-Rac in both wild-type and *Ccr5*^{-/-} cells, indicating that the function was independent from CCR5. However, the effects of these constructs on other markers, such as *Itgαv*, *Mmp3* and *Mmp13*, were specific to *Ccr5*^{-/-} cells, suggesting CCR5-dependent effects. These results are fully described and discussed in the revised manuscript.

Fig 8. In vivo data need to be expanded by adding OC count, and including OC morphology. What are the effects of CCL5 on OB functions? What is the source of CCL5 in vivo and what are the levels of CCL5 in mice?

We have added the morphometric data from the experiments. These data are shown in Figure 7D and F. The phenotypic discrepancies among the anti-CCL5-injected bones, the *Ccl5*^{-/-} bones reported by Wintges et al. in JBMR 2013, and the *Ccr5*^{-/-} bone in the present study are fully addressed in the Discussion section.

In terms of the source of CCL5, we measured the serum levels of CCL3 and CCL5, as shown in Figure 7A in the manuscript. The serum concentration of CCL5 was almost 60 pg/mL, while the level of CCL3 was below the limit of detection. We also conducted immunofluorescence staining of bone sections using anti-CCL5 (shown in Sup. Figure 7B). CCL5 in circulation and the CCL5 expressed by osteoblasts, osteoclasts and endothelial cells were the major sources of CCL5.

Number of mice for in vivo experiments need to be reported.

This is now described in figure legends of Figures 4 and 7.

Reviewers' comments:

Reviewer #1 (Remarks to the Author):

The authors have performed more experiments and have done the proper quantification of the analyses, which improved the Quality of the manuscript.

Reviewer #2 (Remarks to the Author):

I have no further comments

Reviewer #3 (Remarks to the Author):

The authors responded to most of the criticisms raised by the reviewers. However, the revised version of this paper is extremely difficult to digest. There is no cohesive story and the authors often jump from one set of experiments to a different one without any logical explanation, to then go back to add more experiments related to the first part. Furthermore, the result section is lengthy and redundant. Many concepts throughout the manuscript are very convoluted such as in the introduction lines 83-90. Can the authors be more specific? Can they more clearly state what the controversial findings on CCR5 in bone are? Finally, some of the conclusions made on the effects of CCR5 inhibition on osteoclastogenesis are not supported by the data presented.

Specific comments:

The term CCR-5 antagonist should be used instead of Maraviroc.

Line 124 sentence should read "as confirmed by lack of..."

In Fig1 TRAP activity in C (D2-6) should be next to 1B since the authors show the same D2-6 time point.

PTX in 1D show a lot of TRAP positive cells, and so does anti-hCCR5. How can the authors conclude that PTX and anti-CCR5 block OCgenesis when administered from Day2-4? This conclusion is not proven by the data shown. In fact the authors contrast themselves and show that anti-CCR5 treated cells form OCs but they have cytoskeletal defects. Thus, it appears to me that both PTX and anti-CCR5 do not block osteoclastogenesis but impair actin dynamics.

How was the podosome formation in 1F and podosome assembly in 1G examined? Some experimental details need to be added for the reader to understand the data.

Statistics on the expression profile of OC treated with anti-CCR5 need to be provided. Why did the author perform these experiments using a cell line instead of primary cells? U937 is a human leukemic monocyte lymphoma cell line. Data on OC markers must be performed in primary cells.

According to the text, cells exposed to the blocking Ab (dose 1 or 10) have disrupted actin rings (lines 173-176.) But the actin rings shown in Sup Fig1D appear normal prior to exposure to cold PBS (Normal medium). There is discrepancy between the data shown and the conclusion made. Data in Sup Fig1E are not described.

The sentence 186-190 is confusing. If blockade of CCR5 affects OB differentiation, why are there no differences in Sup Fig 2 between ctr Ab and anti-CCR5Ab?

Fig 3. Image in 3B at x20 shows plenty of normal looking actin rings. It would be more helpful if the authors report the number of large and small actin rings per cell

Fig3. The authors refer to Sup Movie 1. In the WT well, two cells are shown. They appear of different size thus the membrane contractions might simply due to a different stage of differentiation. When the large WT cell movie is compared to the KO movie, membrane extensions are not that different.

Data in 3A are hard to interpret and they seem very subjective.

Authors should substitute FAK to Pyk2 in line 225

Data on avb3 expression is confusing. Western blot show similar levels of av but day5 RT-PCR shows significant lower levels in the KO cells. B3 western blot show slightly reduced protein and mRNA levels on day 4, but no differences at day 5. What's the significance for these incremental changes in integrin expression? The only convincing data in this figure are the WB of c-Src and p-Pyk2.

The authors should explain why they analyzed Akt. Is Akt part of the avb3/Src/Pyk2 adhesion complex? A reference should be included.

I would move the Akt WB to the right, together with the other RANKL signaling pathways.

Data in 3F lack the pSrc WB and the avb3/c-Src/Pyk2 complex formation to support the conclusion stated in line 273.

Results in fig4 are repetitive and unnecessary long (4 and half pages!).

Why are the authors reporting in E the number of OCs? This value is meaningless. NOC/BS or NOC/B.Pm is sufficient to demonstrate that the KO mice have higher number of what appear to be dysfunctional OCs.

The description of the in vivo bone phenotype in Fig4 seems to interrupt the flow of the story, thus rendering the following Fig5 very difficult to digest as it goes back to some aspects that were described in Fig3. It is also not clear why CCR5^{-/-} cells, which have impaired adhesion complex, have a similar migratory rate as WT (Sup Fig6). Several data in the literature have shown that the avb3 complex is required for cell migration in response to MCSF.

Figure 5. As ctr for Ab specificity, authors need to report in vivo staining of CCR5^{-/-} OCs with CCR5Ab.

Signaling in WT cells in response to CCL5 is not convincing. While it appears that CCL5 induces FAK activation in cells prior to RANKL treatment, differences seem to be gone following stimulation with RANKL. Quantification done by the authors does not seem to reflect the WB (see line 4 of 60min with RANKL which indicates having 4 fold higher p-FAK levels than line 1 alas 0 RANKL). Similarly, CCL5 does not appear to greatly increase pSrc in Fig 5E. However in 5F IP shows higher pSrc levels. What's the time point used in 5F? And why is there discrepancy between 5E and 5F? In lines 402-404 authors say that CCL5-CCR5 axis cooperatively stimulated Src and Pyk2-mediated signaling through the activation of FAK in cultured osteoclasts. This sentence is not supported by their data.

What's the logic to add another ligand, CCL9?

Another example of disrupted flow is shown in the session describing Fig 6, which comes after figures 5H-J. The latter does not add much to the story and perhaps should be added earlier in the paper to justify why the authors analyzed integrin signaling in the first place.

Fig 6. Why is vav3 shown? I know it's a GTPase, but that should be explained to the reader. It also seem strange that Vav3 is analyzed in the KO cells but not in cells stimulated with CCL5.

It would be much better to combine in the same figure Rac and Rho data in cells treated with CCL5 and in the KO cells.

Data in 6D are not convincing. Are there real differences in integrin expression? And are these differences contributing to the OC phenotype?

The authors reported earlier that the KO cells have defects in actin rings. However they now show in 6E/F no differences in actin ring numbers, but defective resorption. The authors should carefully quantify the actin rings in the wells based on their appearance. Also, the ruffled border of resorbing OCs should be analyzed by EM.

In vivo data should all go together, possibly at the end of the manuscript (bone phenotype of KO mice and following CCL5 administration). It is not clear why the CCR5^{-/-} have no basal phenotype while CCL5 blockade increase

Reviewer #3 (Remarks to the Author):

The authors responded to most of the criticisms raised by the reviewers. However, the revised version of this paper is extremely difficult to digest. There is no cohesive story and the authors often jump from one set of experiments to a different one without any logical explanation, to then go back to add more experiments related to the first part. Furthermore, the result section is lengthy and redundant. Many concepts throughout the manuscript are very convoluted such as in the introduction lines 83-90. Can the authors be more specific? Can they more clearly state what the controversial findings on CCR5 in bone are? Finally, some of the conclusions made on the effects of CCR5 inhibition on osteoclastogenesis are not supported by the data presented.

We wish to express our thanks to the Reviewer #3 for the constructive comments to our revised manuscript. According to this general comment, we have made substantial changes in the revised manuscript. Our *in vitro* observations are first shown in Figures 1-5, and *in vivo* data are in the last part: Figures 6 and 7, so that *in vitro* findings are described in a cohesive manner without any disturbance by *in vivo* findings. We have introduced logical explanations between experiments, and tried to reduce redundant description. The importance of our molecular and cellular findings in CCR5 in osteoclastogenesis are more clearly elucidated in the Introduction section. Some of the conclusions on the effects of CCR5 inhibition, have been more carefully interpreted and described by reflecting specific comments provided by the Reviewer #3. We have conducted the experiments and analyses that were suggested by the Reviewer #3. Our answers to the specific comments are shown below.

Specific comments:

The term CCR-5 antagonist should be used instead of Maraviroc.

“The CCR5-antagonist Maraviroc” is now termed especially in the first appearance in each section.

Line 124 sentence should read “as confirmed by lack of...

This sentence is revised. It reads;

(Line 127) The blockade of functional CCR5 for the duration of the entire differentiation process on days 2-6 (D2-6) clearly inhibited osteoclastic function in a dose-dependent manner, as confirmed by the impaired formation of actin rings and resorption pits on dentin slices (Figure 1B).

In Fig1 TRAP activity in C (D2-6) should be next to 1B since the authors show the same D2-6 time point.

We did not change this part, because 1B shows pictures of the actin rings and pit formation on dentin slices while 1C shows TRAP activity and its statistic comparison,

though TRAP activity in C (D2-6) and 1B show the same time point. The time course on days 2-6 (D2-6) are highlighted in blue in 1A-C.

PTX in 1D show a lot of TRAP positive cells, and so does anti-hCCR5. How can the authors conclude that PTX and anti-CCR5 block OCgenesis when administered from Day2-4? This conclusion is not proven by the data shown. In fact the authors contrast themselves and show that anti-CCR5 treated cells form OCs but they have cytoskeletal defects. Thus, it appears to me that both PTX and anti-CCR5 do not block osteoclastogenesis but impair actin dynamics.

We have agreed this comment. We must admit that we made an over-interpretation of these data. Corresponding sentences, now, read;

(Line 141) The reduced size of TRAP-positive multinucleated cells incubated with anti-hCCR5 neuAb for days 2-4 (D2-4) was also morphologically confirmed on culture dishes (Figure 1D). PTX (Pertussis toxin), an inhibitor of G α i and thus a pan-inhibitor of chemokine receptors, effectively inhibited the actin ring formation, pit formation and TRAP activity, and reduced the size of TRAP-positive cells (Figure 1B, C, D).

How was the podosome formation in 1F and podosome assembly in 1G examined? Some experimental details need to be added for the reader to understand the data.

We have added more explanation in data shown Fig1F-H. Corresponding sentences read;

(Line 149) We then performed structured illumination microscopy (SIM), which allowed for the visualization of the actin-enriched podosomes constituting the actin rings (Figure 1E, F, G, H). SIM demonstrated that the blockade of CCR5 for days 2-6 (D2-6) inhibited both the formation and assembly of podosomes (Figure 1F, G). A quantitative analysis also showed that the size of the podosomes formed by osteoclasts incubated with anti-hCCR5 neuAb for days 2-6 (D2-6) was significantly reduced in comparison to those formed by control cells (Figure 1H). SIM imaging further demonstrated that the actin stress fibers inside the podosome belts in cells treated with anti-hCCR5 neuAb for days 2-6 (D2-6) were disrupted in comparison to control cells (Figure 1F, right panels).

Statistics on the expression profile of OC treated with anti-CCR5 need to be provided. **Statistics on corresponding data shown in Sup. Figure1 have been introduced.**

Why did the author perform these experiments using a cell line instead of primary cells? U937 is a human leukemic monocyte lymphoma cell line. Data on OC markers must be performed in primary cells.

We used U937 for experimental reasons as is described in the revised text.

Corresponding sentences read;

(Line 163) To confirm the functional specificity of the anti-hCCR5 neuAb, we knocked down the CCR5 expression in cultured human osteoclastic cells differentiated from promyelomonocytic U-937 cells³² using a specific siRNA (Sup. Figure 1B). We used U-937 cells for their better gene transfer rate and more feasible expansion of cell

population than those of normal human osteoclast precursors, which enabled us to conduct statistical analyses of TRAP activity and transcriptional levels of osteoclastic markers (Sup. Figure 1B, C).

According to the text, cells exposed to the blocking Ab (dose 1 or 10) have disrupted actin rings (lines 173-176.) But the actin rings shown in Sup Fig1D appear normal prior to exposure to cold PBS (Normal medium). There is discrepancy between the data shown and the conclusion made.

Data in Sup Fig1E are not described.

In fact, acting ring formation on dentin slices is hard to be pictured because dentin slice is not transparent. For this microscopic technical reason, demonstrative pictures with higher magnification and better resolution cannot be acquired. We have indicated disruptive actin rings by white arrows for better demonstration in the revised Sup. Figure 1D.

Data in Sup Fig1E are now described. Corresponding sentence now read; (Line 194) SIM-based observation confirmed the re-assembly of podosomes in normal differentiation medium after osteoclasts on a glass-bottomed dish with either control antibodies or anti-hCCR5 neuAb (10 µg/mL) were subjected to cold PBS treatment (Sup. Figure 1E).

The sentence 186-190 is confusing. If blockade of CCR5 affects OB differentiation, why are there no differences in Sup Fig 2 between ctr Ab and anti-CCR5Ab?

We agreed that this sentence was confusing. Corresponding sentence is now revised as;

(Line 199) The blockade of CCR5 in the differentiation of human osteoblasts (hOBs) from mesenchymal stromal cells (hMSCs) did not cause any significant changes in the mineralization or expression levels of differentiation markers such as RUNX2, SP7/OSTERIX, RANKL and ALP of osteoblasts (Sup. Figure 2A, B).

Fig 3. Image in 3B at x20 shows plenty of normal looking actin rings. It would be more helpful if the authors report the number of large and small actin rings per cell

As mentioned above, demonstration of actin ring formation on dentin slices is somehow troublesome. Therefore, this comment was very helpful. Because definition of large and small actin rings was not easy, we have added scoring of the number of action rings to Figure2B, and revised corresponding sentences as below;

(Line 221) The actin rings in Ccr5^{-/-} osteoclasts were significantly disrupted in comparison to wild-type cells (Figure 2B). Our scoring indicated that the actin rings in Ccr5^{-/-} osteoclasts on dentin slices fractioned and significantly reduced in size, thus significantly increased their number compared to those in wild-type cells.

Fig3. The authors refer to Sup Movie 1. In the WT well, two cells are shown. They appear of difference size thus the membrane contractions might simply due to a different stage of differentiation. When the large WT cell movie is compared to the KO movie, membrane extensions are not that different.

Data in 3A are hard to interpret and they seem very subjective.

As shown in Sup. Figure3A *Ccr5*-deficient osteoclasts in a plastic-bottomed dish is averagely lager in size than wild-type cells. We have added more experimental explanation to the corresponding texts as below;

(Line 242) The impaired podosome assembly in *Ccr5*-deficient osteoclasts prompted us to analyze the movements of osteoclasts derived from *Ccr5*^{-/-} and wild-type mice on culture dishes by time-lapse microscopy (Figure 3A). GFP-expressing osteoclasts were monitored in time-lapse images that were captured at 10 min intervals for 36 h; the images were then statistically analyzed by image processing (n=6 and 9 multinucleated cells were analyzed from *Ccr5*^{-/-} and wild-type bones, respectively). Figure 3A shows representative time-lapse images showing the cellular movements of the *Ccr5*^{-/-} and wild-type osteoclasts (see also Sup. Movie 1), in which cells showing parameters closest to the mean value were selected.

Authors should substitute FAK to Pyk2 in line 225

The corresponding sub-heading have been revised. It reads;

(Line 240) *Ccr5*-deficiency impairs the cellular locomotion of osteoclasts associated with the disarrangement of adhesion complex including Pyk2.

Data on avb3 expression is confusing. Western blot show similar levels of av but day5 RT-PCR shows significant lower levels in the KO cells. B3 western blot show slightly reduced protein and mRNA levels on day 4, but no differences at day 5. What's the significance for these incremental changes in integrin expression? The only convincing data in this figure are the WB of c-*Src* and p-Pyk2.

We agreed with this comment on integrins avb3 expression. Expression of these integrins tended to be reduced in *Ccr5*-deficient cells, at least we have never observed their upregulation in *Ccr5*-deficient cells compared to wild-type cells. However, in fact, our data on integrins $\alpha v\beta 3$ expression shown in previous Figure 3 did not show a cohesive result. Therefore, we have decided that we remove these data and corresponding description from Figure 3.

The authors should explain why they analyzed Akt. Is Akt part of the avb3/*Src*/Pyk2 adhesion complex? A reference should be included. I would move the Akt WB to the right, together with the other RANKL signaling pathways.

As proposed, we have moved the Akt WB to the right panels. Now it is shown with the other RANKL signaling pathways. We have added a reference, Wang MW, et al. J Clin Invest 114, 206-213 (2004).

Data in 3F lack the pSrc WB and the avb3/c-Src/Pyk2 complex formation to support the conclusion stated in line 273.

As proposed, we have added Src and p-Src WB to Figure 3F. Corresponding sentences reads;

(Line 273) The phosphorylation of Pyk2 in osteoclasts is activated by integrin ligands as well as by RANKL stimulation. When osteopontin, a major ligand for integrins in osteoclasts, was coated on culture dishes, RANKL stimulation induced the phosphorylation of Src and markedly enhanced the level of phosphorylated Pyk2 in wild-type cells; however, these activations were also hampered in *Ccr5*^{-/-} osteoclasts (Figure 3D).

Results in fig4 are repetitive and unnecessary long (4 and half pages!).

Why are the authors reporting in E the number of OCs? This value is meaningless. NOC/BS or NOC/B.Pm is sufficient to demonstrate that the KO mice have higher number of what appear to be dysfunctional OCs.

Description of *in vivo* data shown in previous Figure 4 have been reduced to 3 and a half pages. This data set is now shown in Figure 6 as suggested in the next comment.

We do not think showing the number of OCs is meaningless. This value is compatible with other related morphometric values, firmly supporting an important bone phenotype in *Ccr5*-deficient mice. Therefore, we keep this data as well as other morphometric values.

The description of the *in vivo* bone phenotype in Fig4 seems to interrupt the flow of the story, thus rendering the following Fig5 very difficult to digest as it goes back to some aspects that were described in Fig3.

We agreed with this comment on the *in vivo* bone phenotype in Fig4. We have moved the data previously shown in Figure 4 to Figure 6, and accordingly moved the description. Thanks to this comment, the data description flows in a better way.

It is also not clear why *CCR5*^{-/-} cells, which have impaired adhesion complex, have a similar migratory rate as WT (Sup Fig6). Several data in the literature have shown that the avb3 complex is required for cell migration in response to MCSF.

The migration assay shown in previous Sup. Figure 6 was taken by using cells prior to RANKL stimulation. Phenotypic differences in *CCR5*^{-/-} cells become obvious in later stage of osteoclasts, which is discussed in Discussion.

Figure 5. As ctr for Ab specificity, authors need to report in vivo staining of CCR5^{-/-} OCs with CCR5Ab

As suggested, we have added the staining of CCR5^{-/-} OCs with CCR5Ab for confirming the specificity of this Ab. This is an important experiment. This data is shown as Figure 4A in the revised version.

Signaling in WT cells in response to CCL5 is not convincing. While it appears that CCL5 induces FAK activation in cells prior to RANKL treatment, differences seem to be gone following stimulation with RANKL. Quantification done by the authors does not seem to reflect the WB (see line 4 of 60min with RANKL which indicates having 4 fold higher p-FAK levels than line 1 alas 0 RANKL). Similarly, CCL5 does not appear to greatly increase pSrc in Fig 5E. However in 5F IP shows higher pSrc levels. What's the time point used in 5F? And why is there discrepancy between 5E and 5F?

In lines 402-404 authors say that CCL5-CCR5 axis cooperatively stimulated Src and Pyk2-mediated signaling through the activation of FAK in cultured osteoclasts. This sentence is not supported by their data.

We totally agreed with these comments. Accordingly, we have rephrased the data description and our conclusion. As for the WB of p-Src, we explained the differences between E and F in the text. Corresponding sentences now read;

(Line 344) As shown in Figure 4E, rmCCL5 stimulation induced the phosphorylation of FAK, even at 0 min (prior to RANKL treatment). This did not occur after RANKL stimulation without rmCCL5. This elevated level of the phosphorylation of FAK by the pre-incubation with rmCCL5 lasted in 30 minutes after RANKL stimulation. The pre-incubation with rmCCL5 also enhanced the phosphorylation of Pyk2 prior to RANKL treatment. RANKL stimulation induced the phosphorylations of Src at 15-30 minutes and Pyk2 at 15-60 minutes after the stimulation. To confirm the induced phosphorylation of Src by RANKL or the combination of rmCCL5 and RANKL in an enriched fraction of proteins containing phosphorylated tyrosine residues at 15 minutes after RANKL stimulation, we conducted immunoprecipitation with phosphotyrosine antibodies. After the immunoprecipitation, the expression of phosphorylated Src following the combination treatment with RANKL and CCL5 was markedly increased in comparison to the expression observed after treatment with RANKL alone (Figure 4F). These molecular analyses suggested that the CCL5 stimulated FAK-mediated signaling and support RANKL-induced signaling involving Src and Pyk2 in cultured osteoclasts.

These data are shown as Figure 4E-G in the revised version.

What's the logic to add another ligand, CCL9?

We have removed data of CCL9 from the previous Figure 5G of quantification of active forms of Rac and Rho. The revised data is shown as Figure 4G in the revised version.

Another example of disrupted flow is shown in the session describing Fig 6, which comes after figures 5H-J. The latter does not add much to the story and perhaps should be added earlier in the paper to justify why the authors analyzed integrin signaling in the first place.

As stated above, previous Figure 4 of *in vivo* bone phenotype data have been move to Figure 6 in the revised version. Accordingly, the data shown in previous Figure 6 is now in Figure 5, so that this *in vitro* data now follows other *in vitro* data. In fact, the logic flows better, we suppose.

Fig 6. Why is vav3 shown? I know it's a GTPase, but that should be explained to the reader. It also seem strange that Vav3 is analyzed in the KO cells but not in cells stimulated with CCL5.

It would be much better to combine in the same figure Rac and Rho data in cells treated with CCL5 and in the KO cells.

We analyzed Rac and Rho in *Ccr5*-deficient cells because these are common target molecules under integrin and chemokine pathways. Vav3 is an upstream molecule of these small GTPases. In the revised manuscript, we have introduced this logic flow with sentences;

(383) To further investigate how phenotypic signaling was affected by *Ccr5*-deficiency in osteoclastogenesis based on our pathway analysis, we examined the activities of small GTPases as intracellular signaling molecules that are involved both in focal adhesion- and chemokine-mediated signals^{47, 48}. Rho family GTPases are reportedly activated through CCR1- and CCR5-mediated pathway in macrophages⁴⁷. It is established that $\alpha v\beta 3$ integrin activates signaling complex containing Vav3 that regulates cytoskeletal organization of osteoclast⁴⁸. Vav3 is guanine nucleotide exchange factor that is expressed relatively specific to osteoclasts and transits small GTPases of the Rho family from their inactive GDP to their active GTP-bound form. In *Ccr5*^{-/-} osteoclasts, the levels of Vav3 and the phosphorylated forms of FAK (revealed by immunoblotting) were markedly downregulated in comparison to wild-type cells (Figure 5A), suggesting that the signaling of the FAK-Vav3 pathway was impaired.

These data is now shown in Figure 5A.

Data in 6D are not convincing. Are there real differences in integrin expression? And are these differences contributing to the OC phenotype?

We have removed data on *Itgb3* shown in previous Figure 6D, because the expression of *Itgb3* in this experiment did not support the phenotypic rescue of *Ccr5*-deficient osteoclasts by Rac or Rho activation.

The revised data is shown in Figure 5E.

The authors reported earlier that the KO cells have defects in actin rings. However they now show in 6E/F no differences in actin ring numbers, but defective resorption. The authors should carefully quantify the actin rings in the wells based on their appearance. Also, the ruffled border of resorbing OCs should be analyzed by EM.

We have removed the data of actin ring formation on dentin slices shown in previous Figure 6E and 6F in the revised manuscript, because we show similar data on a glass-bottom dish in previous Figure 5C. Another reason is that, as stated above, statistic evaluation of the number of action rings implies technical difficulty.

These data are now shown Figure 5C and 5D, respectively, in the revised version.

In vivo data should all go together, possibly at the end of the manuscript (bone phenotype of KO mice and following CCL5 administration). It is not clear why the CCR5^{-/-} have no basal phenotype while CCL5 blockade increase

We appreciate this comment. Accordingly, in vivo data are shown and described together in the end of the revised version of manuscript. A possible explanation of phenotypic discrepancy between *Ccr5*-deficiency and CCL5 blockade is that the later involve the loss of CCR1 function. Other possibilities together are fully discussed in the Discussion section.

REVIEWERS' COMMENTS:

Reviewer #3 (Remarks to the Author):

The reviewers sufficiently addressed my concerns.